# CURIOSITY-DRIVEN UNSUPERVISED DATA COLLECTION FOR OFFLINE REINFORCEMENT LEARNING

## ABSTRACT

In offline reinforcement learning (RL), while the majority of efforts are focusing on engineering sophisticated learning algorithms given a fixed dataset, very few works have been carried out to improve the dataset quality itself. More importantly, it is even challenging to collect a task-agnostic dataset such that the offline RL agent can learn multiple skills from it. In this paper, we propose a **C**uriosity-driven **U**nsupervised **D**ata **C**ollection (CUDC) method to improve the data collection process. Specifically, we quantify the agent's internal belief to estimate the probability of the $k$-step future states being reachable from the current states. Different from existing approaches that implicitly assume limited feature space with a fixed temporal distance between current and next states, CUDC is capable of adapting how many steps into the future that the dynamics model should predict. Thus, the feature representation can be diversified with the dynamics information. With this adaptive reachability mechanism in place, the agent can navigate itself to collect higher-quality data with curiosity. Empirically, CUDC surpasses existing unsupervised methods in sample efficiency and learning performance in various downstream offline RL tasks of the DeepMind control suite.

## 1 INTRODUCTION

Deep reinforcement learning has demonstrated remarkable breakthroughs in games, robotics, and navigation in complex environments (Kiran et al., 2021; Singh et al., 2022; Sun et al., 2022a). For online RL, agents constantly update the policy to acquire different skills through active interactions with the environments. However, online RL is impractical in many real-world environments as direct interactions with the environments might be expensive or dangerous (Kiran et al., 2021; Singh et al., 2022). In recent years, offline RL has become a promising research area to cope with limited interactions, where agents learn a policy exclusively from previously-collected experiences stored in a fixed dataset (Levine et al., 2020; Kostrikov et al., 2021; Fujimoto & Gu, 2021).

In view of the growing popularity of offline RL, the majority of current research focuses on model-centric practices by successively developing new algorithms (Kumar et al., 2020; Janner et al., 2021; Matsushima et al., 2021; Emmons et al., 2022; Kumar et al., 2022). Despite the rapid progress in these algorithmic advances, their performances are inevitably limited by the quality of the pre-collected dataset itself. Recently, the concept of data-centric approaches has become critical in the machine learning community, emphasizing the importance of improving the training data quality over algorithmic advances (Ng, 2021; Motamedi et al., 2021; Patel et al., 2022). Motivated by this, the offline RL research community attempts to eye on ways of engineering the training data (Prudencio et al., 2022). To focus on more useful data, one solution is to exploit the sample importance with sampling (Zhang et al., 2020) or re-weighting (Wu et al., 2021). Different from this approach, we aim to collect a higher-quality dataset that can be directly used for offline RL agents.

More importantly, it is even more desirable yet challenging to collect a task-agnostic dataset such that offline RL agents are able to extract effective policies for multiple downstream tasks. To analyze and understand these challenges, ExORL (Yarats et al., 2022) empirically shows that unsupervised RL methods are superior to supervised methods to collect the exploratory data that allows even vanilla off-policy RL algorithms to effectively learn offline and acquire different skills. Nevertheless, these existing methods pre-define a fixed temporal distance between current states and future states to train the models, which implicitly limits the diversity in the learned feature representation.

As we observe that fixing the temporal distance between current and future states may limit the feature space and result in low-quality dataset, it is desired to enhance feature representation by exploiting the reachability from current state to more distant future states. Although existing works of reachability analysis have been introduced in RL (Savinov et al., 2019; Péré et al., 2018; Ivanovic et al., 2019; Yu et al., 2022), these approaches are however not directly applicable. For example, Savinov et al. (2019) only considers the reachability in a binary case, and it requires extensive comparisons to the stored embeddings in memory. Moreover, the reachability in goal space exploration requires kernel density estimation, increasing the computational cost substantially. Different from these approaches, we propose a **C**uriosity-driven **U**nsupervised **D**ata **C**ollection (CUDC) method with a novel reachability module. Inspired by the fact that human curiosity can enhance the learning process by motivating human beings on the novel knowledge that is beyond ones' perception (Rossing & Long, 1981; Markey & Loewenstein, 2014; Sun et al., 2022b), CUDC facilitates the agent to collect a dataset curiously without any task-specific reward. In particular, we define the reachability module to characterize the probability of a $k$-step future state being reachable from the current state, with no episodic memory or feature space density modeling required. This module allows the agent to automatically determine how many steps into the future that the dynamics model should predict, where the learned feature representation can be incorporated with the information of dynamics. Compared with the existing unsupervised methods, it avoids relying on the fixed feature space by gradually expanding to more distant future states. With the enhanced representation learning, a mixed intrinsic reward encourages curious exploration towards more meaningful state-action space as well as the under-learned states. As a result, the collected dataset can lead to improved sample efficiency and better performances in downstream offline RL tasks.

Our contributions can be summarized as follows. 1) We are the first to introduce reachability for improving data collection in offline RL, which is defined in a more efficient way and can enable the agent to navigate curiosity-driven learning coherently. 2) We empirically show that adapting the number of steps between current and future states to perform increasingly challenging prediction can enhance feature representation with information of the dynamics, thereby improving the collected dataset quality. 3) With the learned state and action representations, CUDC additionally incentivizes the agent to explore diverse state-action space as well as the under-learned states with high prediction errors through a mixed intrinsic reward and regularization. 4) Under the ExORL (Yarats et al., 2022) setting, CUDC outperforms the other unsupervised methods to collect the dataset that can be learned offline in multiple downstream tasks of the DeepMind control suite (Tassa et al., 2018).

## 2 RELATED WORKS

**Reachability in RL**   Savinov et al. (2019) constructed a reachability network to estimate how many environment steps to take for reaching a particular state. It intrinsically rewards the agent to explore the state that is unreachable i.e., takes more than a fixed threshold step, from other states in memory. However, it only takes the binary case of reachability into consideration and is quite inefficient when comparing the similarity with all stored states in memory. In the goal exploration tasks, Péré et al. (2018) defined the reachability of a goal with an estimated density and proposed to sample increasingly difficult goals to reach during exploration. Although the goal space can be learned in an unsupervised manner other than in a specifically engineered way, its sampling process requires a kernel density estimator, increasing the computational cost substantially. Following the similar idea, BARC (Ivanovic et al., 2019) adapts the initial state distribution gradually from easy-to-reach to challenging-to-reach goals. As a result, agents can perform well even in a hard robotic control task. Recently, RCRL (Yu et al., 2022) has shown that leveraging reachability analysis (Hsu et al., 2021) can help learn an optimal safe policy by expanding the limited conservative feasible set to the largest feasible set of the state space.

**Curiosity-Driven RL**   Curiosity-driven RL intrinsically encourages agents to explore the task environment in a human-like way, which is of vital importance when the task-specific rewards are sparse or absent (Aubret et al., 2019; Sun et al., 2022b). The main type of curiosity-driven RL is to incorporate an intrinsic reward that self-motivates agents to explore based on various aspects of the state, such as novelty (Bellemare et al., 2016), entropy (Seo et al., 2021; Liu & Abbeel, 2021b), reachability (Savinov et al., 2019), prediction errors (Pathak et al., 2017; Berseth et al., 2020), complexity (Campero et al., 2020), and uncertainty (Pathak et al., 2019; Sekar et al., 2020; Li et al.,

2021). Another type of curiosity-driven RL is to prioritize the experience replay towards under-explored states (Schaul et al., 2016; Zhao & Tresp, 2019; Brittain et al., 2019; Jiang et al., 2021). However, the curiosity mechanism can be further exploited and introduced into other components of RL for exploration, as shown in CCLF (Sun et al., 2022a). Therefore, we propose to curiously adapt the temporal distance to explore more distant future states, enhancing the learned representation space with dynamics information. Meanwhile, our method regularizes the Q-learning by assigning the importance weights through a curiosity weight to focus more on under-learned tuples.

**Unsupervised Data Collection** ExORL (Yarats et al., 2022) evaluates 9 unsupervised data collection algorithms, demonstrating a superior capability over supervised methods for multi-task offline learning. In particular, knowledge-driven models of ICM (Pathak et al., 2017), Disagreement (Pathak et al., 2019), and RND (Burda et al., 2019) encourage agents to explore by maximizing the prediction errors of the states. Data-driven models of APT (Liu & Abbeel, 2021b) and ProtoRL (Yarats et al., 2021) incentivize to uniformly explore the entire state space. By leveraging some prior information, competence-based models of DIAYN (Eysenbach et al., 2019), SMM (Lee et al., 2019) and APS (Liu & Abbeel, 2021a) encourage agents to learn diverse skills. However, all these methods are not tailored for data collection; instead, they were originally proposed for online pretraining and fine-turning at task learning process as evaluated in the URLB benchmark (Laskin et al., 2021). Moreover, their feature space is limited by relying on a fixed temporal distance $k$ between the current state and future state for model training. Concurrently, Explore2Offline (Lambert et al., 2022) leverages the intrinsic model predictive control to simulate trajectories and their resulting predicted intrinsic rewards, but it does not consider the sample importance while updating the policy.

## 3 CURIOSITY-DRIVEN UNSUPERVISED DATA COLLECTION (CUDC)

### 3.1 PROBLEM SETTING

We consider the problem of multi-task offline learning with three main steps of data collection, reward relabeling, and downstream offline learning as described in both ExORL(Yarats et al., 2022) and Explore2Offline (Lambert et al., 2022). In the data collection phase, the exploratory agent (data collector) has the access to a Markov Decision Process (MDP) environment with a state of the environment $s \in \mathcal{S}$, an action $a \in \mathcal{A}$ based on a policy $\pi(s)$, a transition probability $p(s'|s,a)$ mapping from the current state $s$ and action $a$ to the next state $s'$, a reward $r$, and a discount factor $\gamma \in [0,1)$ weighting future rewards. In particular, the exploratory agent online interacts with the environment and stores the unlabeled tuples $(s,a,s')$ in the dataset $\mathcal{D}$. The second phase is to relabel the collected dataset $\mathcal{D}$ using the given reward function about the downstream task $r(s,a)$ for each tuple. It transfers information from task-agnostic exploration to downstream tasks. The last step is to perform multiple downstream tasks with an offline RL agent on the labeled dataset. In this paper, we focus on the most challenging part of this problem setting, which is the task-agnostic data collection and we evaluate the quality of the collected dataset in multiple downstream tasks.

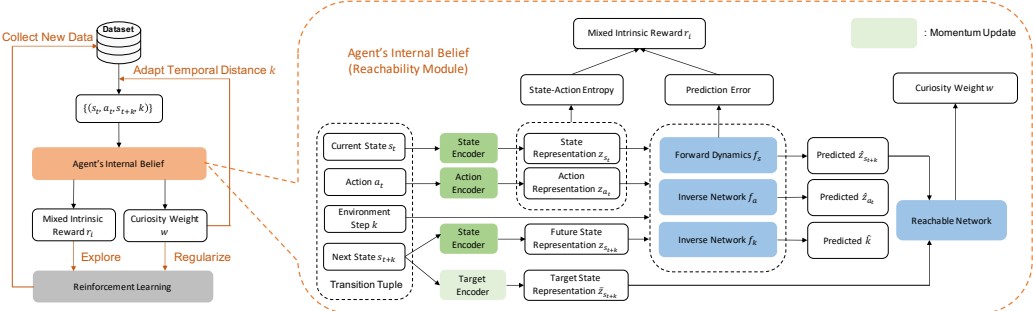

Figure 1: Curiosity-driven Unsupervised Data Collection (CUDC): The diagram on the left illustrates the framework of CUDC. A batch of transition tuples is sampled from the accumulated dataset, and then the reachability between the $k$-step future and current states is measured by agent's internal belief. The reachability module outputs the mixed intrinsic reward to encourage diverse exploration. Meanwhile, it also outputs a curiosity weight to adapt the temporal distance and regularize the backbone RL. Therefore, the agent can collect new data until the capacity is reached. The diagram on the right shows how the agent measures and updates its internal belief on the probability of the $k$-step future states being reachable from the current states.

## 3.2 FRAMEWORK OVERVIEW

In this section, we propose a **C**uriosity-driven **U**nsupervised **D**ata **C**ollection (CUDC) method, which employs DDPG (Lillicrap et al., 2015) as the base RL algorithm for the exploratory agent. As shown in Figure 1 (right), a novel reachability module is constructed to measure the probability of a $k$-step future state being reachable from the current state. With this module in place, the exploratory agent is encouraged to diversely explore by a mixed intrinsic reward, and meanwhile regularize the critic-actor update to focus more on under-learned tuples. Most importantly, the temporal distance of $k$-step between current and future states is adaptively increased to incorporate the dynamics information in the learned feature representation. As a result, the exploration and data collection become more diverse. More details are listed in Algorithm 1.

## 3.3 THE REACHABILITY MODULE

In ExORL(Yarats et al., 2022), the existing unsupervised methods assume the limited feature space by implicitly fixing the temporal distance $k$ between current and future states to train the model. To diversity the feature representation, a natural choice is to employ the reachability analysis that can adjust $k$ during exploration. However, the existing implementations of reachability are not desired due to the limited capability by binary classification of reachable states (Savinov et al., 2019), or costly density estimation of goal space (Péré et al., 2018). As these approaches are not practical in the offline RL setting, we define the reachability in CUDC by the self-supervised estimation on the probability of a $k$-step future state $s_{t_i+k}$ being reachable from the current state $s_{t_i}$, with no expensive density estimation or manual labelling required. As a result, the proposed method can enhance feature representation when expanding the feature space through an adaptive $k$-step. In fact, this motivation has been shown effective in the other works of reachability, such as constrained RL (Yu et al., 2022) and robotics (Ivanovic et al., 2019).

Given a batch of unlabeled tuples $(s_{t_i}, a_{t_i}, s_{t_i+k}, k)_{i=1}^{n}$, we first encode the state features $z_{s_{t_i}} = \phi_s(s_{t_i})$, $z_{s_{t_i+k}} = \phi_s(s_{t_i+k})$ and action feature $z_{a_{t_i}} = \phi_a(a_{t_i})$ with a state encoder $\phi_s(\cdot)$ and an action encoder $\phi_a(\cdot)$, and perform the one-hot encoding for the temporal distance $k$. To facilitate the reachability analysis, a forward dynamic network $\hat{z}_{s_{t_i+k}} = f_s(z_{s_{t_i}}, z_{a_{t_i}}, k; \theta_s)$ is constructed, which takes the inputs of $z_{s_{t_i}}$, $z_{a_{t_i}}$, and the encoded $k$ to predict the future state feature $\hat{z}_{s_{t_i+k}}$, fully exploiting the information of dynamics. It can be updated by an $l_2$ norm loss of $||z_{s_{t_i+k}} - \hat{z}_{s_{t_i+k}}||_2$.

In addition, we want to enforce $\hat{z}_{s_{t_i+k}}$ to match with its own $z_{s_{t_i+k}}$ as much as possible, while keeping apart from the other future states within the same batch. This contrastive intuition is that each future state should be most reachable from its own current state, which can quantify the reachability in a simple and efficient way. In fact, the self-supervised contrastive learning has been recently introduced in RL and is shown to be capable of learning rich representation with more semantic latents (Srinivas et al., 2020; Liu & Abbeel, 2021b). Following this intuition, we can estimate the probability of $s_{t_i+k}$ being reachable from $s_{t_i}$ by

$$l_i = \frac{\exp(h(\hat{z}_{s_{t_i+k}})^T W \bar{h}(\bar{z}_{s_{t_i+k}}))}{\exp(h(\hat{z}_{s_{t_i+k}})^T W \bar{h}(\bar{z}_{s_{t_i+k}})) + \sum_{j=1, j \neq i}^{n} \exp(h(\hat{z}_{s_{t_i+k}})^T W \bar{h}(\bar{z}_{s_{t_j+k}}))} \tag{1}$$

where $n$ is the batch size, $h(\cdot)$ is a deterministic projection function, $W$ is a hidden weight to compute the similarity between the two projections, and $\bar{h}(\cdot)$ as well as $\bar{z}(\cdot)$ are respectively the momentum-based moving average of the projection and state feature to ensure consistency and stability (He et al., 2020). In this way, this reachability network can be updated by the contrastive loss of $\mathcal{L}_{\text{reach}} = -\sum_{i=1}^{n} \log l_i$ in a self-supervised way without maunal labeling.

To further improve the representation learning, the reachability module also includes two inverse models for predicting action feature $\hat{z}_{a_{t_i}}$ and temporal distance $\hat{k}$. Similar to ICM (Pathak et al., 2017) and Disagreement (Pathak et al., 2019), we define $\hat{z}_{a_{t_i}} = f_a(z_{s_{t_i}}, z_{s_{t_i+k}}, k; \theta_a)$ with a backward loss of $||z_{a_{t_i}} - \hat{z}_{a_{t_i}}||_2$, which ensures the encoded features are robust to environment variations that are uncontrollable by the agent. For the inverse model of the $k$-step, $\hat{k} = f_k(z_{s_{t_i}}, z_{s_{t_i+k}}; \theta_k)$ characterizes the prediction with a distribution $\mathbb{P}(k)$. It is updated through a cross-entropy loss, which enables the encoders to capture the dynamics information in the encoded features.

The agent can therefore update its internal belief in a self-supervised manner without any expensive labeling process compared to supervised learning. More importantly, we can allow the temporal distance of $k$-step to adapt during learning, rather than relying on a fixed hyperparameter in many existing unsupervised methods. It is of vital importance as the feature representations of both states and actions become more informative and robust when adjusting the temporal distance of $k$-step. Meanwhile, this module is also capable of computing a curiosity weight $w_i$ for each tuple $i$ as

$$w_i = 1 - l_i \in [0, 1]. \tag{2}$$

Intuitively, a large value of $w_i$ means that the agent mistakenly believes the true future state is unreachable from the current state, which induces high curiosity due to the conflict with agent's current internal belief. It further indicates that this under-learned transition tuple contains novel information, and the encoders are not capable of extracting meaningful features yet. With this reachability module in place, we can enable the agent to perform the task-agnostic dataset collection in a curious manner. Detailed curiosity-driven learning is illustrated in the next subsection.

### 3.4 CURIOSITY-DRIVEN LEARNING

It should be noted that reachability in previous works is only incorporated as an intrinsic reward to encourage diverse exploration. In contrast, our proposed CUDC extends the reachability to more stages of learning. Firstly, it adapts the temporal distance, i.e. $k$-step to explore, enhancing feature representation with the dynamics information of more distant future states. Secondly, it incorporates a mixed intrinsic reward to encourage effective exploration with the expanded feature space. Lastly, it regularizes the critic-actor update for the DDPG algorithm by utilizing the curiosity weights to focus more on under-learned tuples. Different from the 8 existing methods evaluated in ExORL that only utilize intrinsic rewards as curiosity, our CUDC extends curiosity-driven learning to different RL components, improving the data collection process coherently.

#### 3.4.1 ENHANCE FEATURE REPRESENTATION WITH ADAPTIVE $k$-STEP

It is worth noting that all 8 methods discussed in ExORL limit the autonomy of feature space by expecting the agent to reach the future states in exactly 3 steps. Recent works on online pre-training RL such as SPR (Schwarzer et al., 2020) and SGI (Schwarzer et al., 2021) predict agent's own latent state representations multiple steps into the future, addressing the challenge of sample efficiency. However, this approach requires iterative predictions by calling the forward dynamic network for $k$ times. In contrast, CUDC can automatically adjust the temporal distance $k$ to directly perform the $k$-step future state estimation in the proposed reachability module, without increasing the computational complexity substantially. To expand the feature space by leveraging the dynamics information, our main idea is to keep the reachability estimation increasingly challenging with an adaptive $k$-step.

In particular, we adaptively adjust the $k$ by encouraging curiosity, i.e., $k$ is increased by 1 if the agent is not curious enough for the current reachability analysis. For this purpose, we set a threshold $C_w$ for low curiosity and a threshold $C_k$ for the proportion of tuples with low curiosity. Thus, the agent will adapt $k$ when $\frac{\sum_i^n \mathbf{1}[w_i < C_w]}{n} > C_k$. The intuition is that the agent should be encouraged to explore further based on the reachability, when they are no longer curious about the majority of tuples within the same batch. As a result, the feature representation can be enhanced by learning the dynamics of more distant future states. With the expanded feature space, the data collection is performed towards more informative and diverse features. It should be noted that there are many possible ways to vary the $k$-step such as by sampling from a probabilistic distribution. For this reason, an ablation study is also conducted to validate the effectiveness of the proposed curiosity-driven method, compared to other sampling-based methods.

#### 3.4.2 INCORPORATE A MIXED INTRINSIC REWARD

CUDC employs a mixed intrinsic reward of state-action entropy and prediction error of future states. On one hand, the existing methods of APT (Liu & Abbeel, 2021b) and RE3 (Seo et al., 2021) have shown that agents can be encouraged to explore the state space more uniformly, by utilizing the particle-based k-nearest neighbors state entropy as an intrinsic reward. This is consistent with the entropy maximization principle (Beirlant et al., 1997; Singh et al., 2003). However, we believe that the agent should diversely explore not only the state space, but also the action space. Therefore,

---

**Algorithm 1** Implementation of the proposed CUDC

---

**Initialize** parameters of encoders $\phi_s$ and $\phi_a$, forward dynamic network $f_s$, inverse models $f_a$ and $f_k$, projection $h$, critic $Q$, policy $\pi$, hidden weight $W$, and the temporal distance $k$
**Initialize** the task-agnostic dataset $\mathcal{D} = \emptyset$
  **for** each time step $t$ **do**
    // COLLECT TRANSITIONS
    Interact with the environment using the policy $a_t \sim \pi(s_t)$ and observe $s_{t+1}$
    $\mathcal{D} \cup (s_t, a_t, s_{t+1}) \rightarrow \mathcal{D}$
    // UPDATE INTERNAL BELIEF OF REACHABILITY
    Sample a minibatch $\{(s_{t_i}, a_{t_i}, s_{t_i+k}, k)\}_{i=1}^n \sim \mathcal{D}$
    **for** each tuple $i$ in the minibatch **do**
      Encode the state and action, and predict the $t_i + k$'s future state feature $\hat{z}_{s_{t_i+k}}$
      Evaluate the curiosity weight $w_i$ according to Equation (2)
      Compute the intrinsic reward $r_i$ using $w_i$ by Equation (3)
    **end for**
    Update the internal belief of reachability module
    //ADAPT THE K-STEP TO PREDICT
    **if** $\frac{1}{n}\sum_i^n \mathbf{1}[w_i < C_w] > C_k$ **then**
      Increase the temporal distance by $k = k + 1$
    **end if**
    //REGULARIZE CRITIC-ACTOR UPDATE
    Update the critic $Q$ with regularization by Equation (4)
    Update the actor $\pi$ with regularization
    Perform the momentum update for $\bar{h}$ and $\bar{z}$
  **end for**

---

CUDC extends state embedding to state-action embedding and we prove that the k-nearest neighbor entropy estimation in the state-action representation space can be used for entropy maximization.

**Lemma 3.1.** *Let $u = (z_s, z_a)$ represent the state-action representation. The particle-based entropy $\mathcal{H}(u)$ is proportional to a K-nearest neighbor (K-NN) distance,*

$$\mathcal{H}(u) \propto \sum_{i=1}^n \log ||u_i - u_i^{K\text{-}NN}||_2.$$

Following the same idea to treat each tuple as a particle (Liu & Abbeel, 2021b; Seo et al., 2021), we formulate an intrinsic reward to estimate the particle-based entropy, $r_{\mathcal{H}}(s_{t_i}, a_{t_i}) = \log(\frac{1}{N_K} \sum ||u_i - u_i^{K\text{-}NN}||_2 + 1)$ where $u_i = (\phi_s(s_{t_i}), (\phi_a(a_{t_i}))$ and $N_K$ is the number of K-NN. As the encoded features are constantly updated to capture the dynamics of more distant future states in the reachability module, the proposed $r_{\mathcal{H}}$ can lead to a more diverse state-action space exploration, where the state-of-the-art off-policy RL algorithm SAC (Haarnoja et al., 2018) has also shown that exploration through a discounted maximum-policy (actor)-entropy term is more effective.

On the other hand, the prediction errors of future states are often employed as the intrinsic reward to incentivize the agent to explore surprising states that are beyond their exceptions (Pathak et al., 2017; Burda et al., 2018). CUDC additionally integrates the prediction error $r_{\mathcal{E}}(s_{t_i}, a_{t_i}) = ||z_{s_{t_i+k}} - \hat{z}_{s_{t_i+k}}||_2$ in the intrinsic reward as well, where the reachability module is conveniently re-used without additional networks. Finally, the mixed intrinsic reward in CUDC is given by

$$r_i(s_{t_i}, a_{t_i}) = r_{\mathcal{H}}(s_{t_i}, a_{t_i}) + \alpha r_{\mathcal{E}}(s_{t_i}, a_{t_i}) + \beta, \tag{3}$$

where $\alpha$ controls the prioritization of under-learned state exploration and $\beta$ is a constant for numerical stability.

### 3.4.3 REGULARIZE THE CRITIC-ACTOR UPDATE

Furthermore, CUDC adaptively regularizes the backbone DDPG algorithm to focus more on under-learned tuples, by utilizing the curiosity weight $w_i$. In particular, $w = (w_1, w_2, \cdots, w_n)$ quantitatively describes the curiosity weight on each transition tuple, which can be seamlessly used to

characterize the sample importance and regularize both critic and actor updates. Therefore, we can perform the Q-learning in DDPG by minimizing the following objective,

$$\mathbb{E}_{(s,a,s')\sim\mathcal{D}}\left[w\left(Q(s,a)-(r_i(s,a)+\gamma Q_{\text{target}}(s',\pi(s')))\right)^2\right]. \tag{4}$$

Meanwhile, the policy can be updated by maximizing $\mathbb{E}_{(s,a,s')\sim\mathcal{D}}\left[wQ(s,\pi(s))\right]$. In this way, CUDC facilitates the agent to adapt the learning process in a self-supervised manner in the sense that the regularization is controlled by the conceptualized curiosity to exploit sample importance.

## 4 EXPERIMENTAL RESULTS

**Environments**  We evaluate the proposed CUDC on continuous control tasks with state observations in the DeepMind control suite (Tassa et al., 2018), which consists of 12 downstream tasks across 3 main challenging domains: Walker, Quadruped, and Jaco Arm. Walker is a controllable entity with locomotion-related balancing controls, where it can learn walking, running, flipping and standing. Quadruped is a passively stable body in a more challenging 3D environment to learn various locomotion skills of walking, running, standing, and jumping. Jaco Arm is 6 degree-of-freedom robotic arm with a three-finger gripper for object manipulation, where the downstream tasks require it to reach different positions. Note that PointMass Maze is not included since most baseline methods in ExORL have already demonstrated excellent performances.

**Baseline Models**  We compare CUDC against state-of-the-art unsupervised methods across all three categories as benchmarked in ExORL, i.e., a knowledge-driven baseline of ICM (Pathak et al., 2017), data-driven baselines of APT (Liu & Abbeel, 2021b) and ProtoRL (Yarats et al., 2021), and a competence-driven baseline of APS (Liu & Abbeel, 2021a). Meanwhile, a random data collector is also included, which collects the data by performing randomly sampled actions. The other four methods discussed in ExORL are excluded since their performance are less competitive. We set the same hyperparameters and model architecture as reported in ExORL. To demonstrate that all proposed components play important roles in the performance, we also compare four versions of CUDC as follows. $\text{CUDC}_{\text{vary}}^{\text{ICM}}$ and $\text{CUDC}_{\text{vary}}^{\text{APT}}$: adapting the temporal distance of $k$-step by the intrinsic rewards based on the original ICM and APT methods. $\text{CUDC}_{\text{reward}}$: extending to state-action entropy with a mixed intrinsic reward based on $\text{CUDC}_{\text{vary}}^{\text{APT}}$. $\text{CUDC}_{\text{reach}}$: adding the full reachability module without regularization based on $\text{CUDC}_{\text{reward}}$. The detailed implementation and differences from the full model are summarized in Appendix A.

**Model Training and Evaluation**  To impose model stability during learning, we restrict the $k$-step to be increased from 3 to 6 and set the upper and lower bounds for the regularization weights to ensure stability. For further details regarding the network implementation and hyperparameter setting of the proposed CUDC, readers can refer to Appendix A. During data collection, all methods are trained using a DDPG (Lillicrap et al., 2015) agent as the backbone for fairness. They interact with 3 domain environments in the absence of extrinsic rewards for 1M steps. For the main results, a total of 90 datasets (6 algorithms × 3 main tasks × 5 seeds) are collected. After that, relabeling is performed for each downstream task on the collected dataset. During the evaluation, a TD3 (Fujimoto et al., 2018) agent learns offline from each relabeled dataset for 500K steps. We report the performance score at 100K step for sample efficiency and at 500K step for learning performance.

**Main Results on 12 Downstream Tasks**  As shown in Figure 2, no single baseline method can greatly improve the dataset quality for all domains. In contrast, the dataset collected by CUDC is capable of boosting the offline agent's learning performances at 500K step in all 12 downstream tasks across 3 challenging domains, as highlighted in Table 1. Specifically, CUDC outperforms the competence-based method (APS) in the Walker domain by 6%, outperforms the data-based method (APT) in the Quadruped domain by 51%, and outperforms the knowledge-based method (ICM) in the Jaco Arm domain by 10%. As for the sample efficiency, it can be observed in Figure 2 that CUDC leads to the best sample efficiency with a significant margin on 3 downstream tasks of Quadruped. In the easy domain of Walker, CUDC can help the offline agent to converge faster in 3 downstream tasks. However, the sample efficiency in the hardest domain of Jaco Arm is not desirable. It might be caused by too much complexity in this most challenging environment, increasing the difficulty of reachablility analysis. More results are included in Appendix C.1 and evaluation using another offline RL algorithm of CQL (Kumar et al., 2020) is conducted in Appendix C.2.

Table 1: Main results of the offline RL agent on 12 downstream tasks across 3 main domains. The proposed CUDC collects a more useful dataset such that offline RL agents can improve sample efficiency (100K) in 9 out of 12 downstream tasks and achieve better learning performance (500K) in all 12 downstream tasks.

| 100K Step Score | Random | APS | ProtoRL | ICM | APT | CUDC |
|---|---|---|---|---|---|---|
| Walker, Walk | 190±153 | 652±227 | 532±332 | 503±258 | 548±338 | **827±64** |
| Walker, Flip | 175±136 | 590±77 | 610±95 | 530±59 | 579±51 | **615±66** |
| Walker, Run | 53±25 | 368±77 | 332±81 | 233±111 | 372±40 | **381±37** |
| Walker, Stand | 401±295 | 923±76 | 831±318 | 797±312 | 878±101 | **984±11** |
| Quadruped, Walk | 135±101 | 206±25 | 153±119 | **338±211** | 248±22 | **338±147** |
| Quadruped, Run | 145±96 | 183±17 | 121±31 | 210±63 | 249±24 | **256±133** |
| Quadruped, Stand | 271±105 | 334±148 | 193±78 | 585±301 | 524±153 | **618±311** |
| Quadruped, Jump | 223±60 | 237±83 | 150±89 | 466±208 | 391±127 | **483±222** |
| Jaco Arm, Reach Top Left | 5±4 | 63±40 | 68±49 | **88±78** | 42±77 | 54±60 |
| Jaco Arm, Reach Top Right | 49±37 | **119±64** | 76±39 | 99±73 | 51±67 | 32±59 |
| Jaco Arm, Reach Bottom Left | 35±31 | 87±74 | 76±75 | **101±46** | 30±43 | 74±89 |
| Jaco Arm, Reach Bottom Right | 49±46 | 100±75 | 85±66 | 113±89 | 92±42 | **121±79** |
| 500K Step Score | Random | APS | ProtoRL | ICM | APT | CUDC |
| Walker, Walk | 198±266 | 845±38 | 826±67 | 802±67 | 799±73 | **893±34** |
| Walker, Flip | 303±195 | 561±121 | 645±150 | 534±218 | 591±77 | **717±41** |
| Walker, Run | 62±24 | 369±33 | 386±38 | 261±142 | 384±20 | **393±11** |
| Walker, Stand | 519±312 | 949±24 | 954±17 | 868±73 | 890±139 | **971±7** |
| Quadruped, Walk | 77±53 | 169±38 | 177±181 | 231±107 | 363±141 | **425±76** |
| Quadruped, Run | 102±52 | 179±52 | 77±36 | 165±89 | 198±57 | **349±80** |
| Quadruped, Stand | 162±96 | 335±80 | 127±111 | 343±133 | 464±144 | **637±188** |
| Quadruped, Jump | 152±74 | 261±71 | 99±51 | 242±108 | 329±85 | **574±74** |
| Jaco Arm, Reach Top Left | 59±75 | 129±26 | 138±52 | 166±54 | 41±29 | **212±11** |
| Jaco Arm, Reach Top Right | 81±54 | 152±82 | 166±19 | 195±36 | 95±32 | **214±17** |
| Jaco Arm, Reach Bottom Left | 91±68 | 103±74 | 100±68 | **216±16** | 101±56 | **218±7** |
| Jaco Arm, Reach Bottom Right | 107±56 | 197±33 | 149±69 | **229±8** | 131±33 | **229±6** |

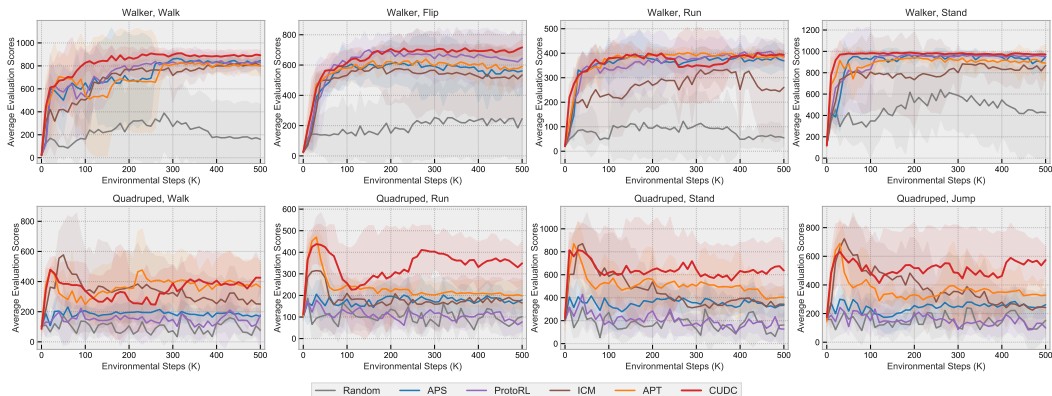

Figure 2: Learning curves of the offline RL agent on the task-agnostic dataset collected by different methods. The proposed CUDC demonstrates superior capability of improving the sample efficiency and learning performances of the offline RL agent.

**Effects of Adapting the $k$-Step** We empirically show that adapting the temporal distance to explore more distant future states can certainly help to enhance the feature representation, and thereby improve the data collection process. By comparing the results in Figure 3, $\text{CUDC}_{\text{vary}}^{\text{ICM}}$ has outperformed ICM significantly, with on average $1.25 \times$ sample efficiency at 100K step and $1.16 \times$ offline learning performance at 500K step. Similarly, $\text{CUDC}_{\text{vary}}^{\text{APT}}$ obtains respectively $1.12 \times$ and $1.04 \times$ scores at 100K and 500K steps across 4 downstream tasks, compared with APT. We note that the standard deviation increases a bit, which might be caused by the introduced complexity of more distant future state when improving the learned representation. Thus, it is desirable to find an adaptive way to smooth this process, e.g., by incorporating the other proposed components inherently.

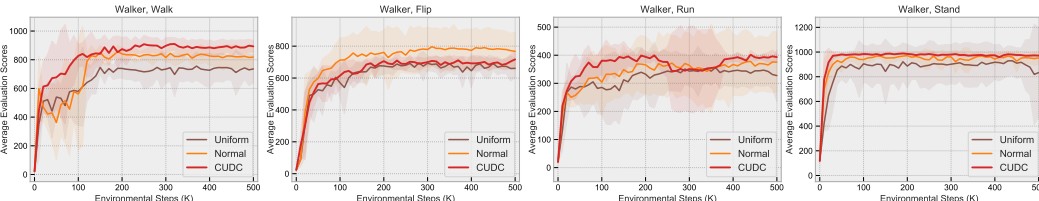

Figure 3: The performance score evaluated at 100K and 500K steps in 4 downstream tasks of Walker. All four versions of CUDC perform better than ICM and APT.

**Effectiveness of the Other Proposed Components**    We additionally incorporate the mixed intrinsic reward to $CUDC_{vary}^{APT}$ as $CUDC_{reward}$, and it further improves both sample-efficiency and learning capability of the offline RL agent, shown in Table 4 and Figure 3. As the new intrinsic reward encourages a more uniform exploration on the state-action space and focuses more on under learned states, the performance at 100K is inevitably unstable, with a 67% increase of standard deviation. Therefore, it is necessary to leverage the proposed reachability module as agent's internal belief and facilitate the data collection. By comparing $CUDC_{reach}$ with $CUDC_{reward}$, the dataset collected by $CUDC_{reach}$ stabilizes the offline learning process by reducing 48% and 25% of standard deviation at 100K and 500K steps. However, its performance scores slightly decrease in 2 tasks. To address this problem, the full model further regularizes the critic-actor update with a curiosity weight, compared with $CUDC_{reach}$. It achieves the best performances with minimum standard deviation at 500K step. More experiments are conducted in Appendix C.4 by removing each proposed components. It can be concluded that all components play important roles to collect a useful dataset for improving the downstream task learning and resolving instability on offline RL agents.

Figure 4: Learning curves of the offline RL agent on 4 downstream tasks of Walker. The $k$-step adaptation proposed in CUDC performs better than the other two sampling methods in 3 out of 4 downstream tasks.

**Adjusting the $k$-Step in Different Ways**    One may wonder how the feature representation improves with different ways of adjusting the temporal distance $k$ in the reachability module. Thus, we explicitly conduct an ablation study on the Walker domain by sampling $k$ uniformly (Uniform) from 3 to 6 and normally (Normal) with an increasing mean. It can be observed in Figure 4 that Uniform performs the worst in all 4 tasks as it cannot adapt the temporal distance in a curious way to enhance the representation learning. At 500K step, it only achieves 85% overall learning capability with 300% increase in the standard deviation, compared to CUDC. Normal to some extent adapts $k$ through an increasing mean, and it even outperforms CUDC in the Flip task. However, its overall performance is still 4.5% weaker than CUDC and its standard deviation is 128% higher than CUDC, indicating the instability issue. Overall, the curious adaptation method proposed in CUDC is the best and it is interesting to investigate more adaptive ways in the future.

## 5    CONCLUSION

We propose CUDC, a curiosity-driven unsupervised data collection method to improve the dataset quality for offline RL agents in the multi-task setting. A reachability module is introduced, quantifying agent's internal belief of estimating the probability of a $k$-step future state being reachable form the current state. Substantially, CUDC can enhance feature representation by adaptively allowing the agent to explore more distant future states in the reachability module. Empirically, CUDC demonstrates superior data collection capability with improved sample efficiency and better performances in downstream multi-task offline learning. We also present this work with insightful empirical evidence of effective data collection methods for future research.

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

## A  CUDC Implementation and Hyperparameter Setting

### A.1  Implementation Details

#### A.1.1  CUDC

In the reachability module of CUDC shown in Figure 1, the state encoder $\phi_s$ is a 1-layer MLP with the ReLU activation. Subsequently, the output is passed to a single normalized fully-connected layer by LayerNorm (Lei Ba et al., 2016) with the tanh nonlinearity applied at the end. The target encoder is momentum updated from the state encoder to obtain $\bar{z}$. The action encoder $\phi_a$ is a 3-layer MLP with the ReLU activation. For the forward network $f_s$ and the backward networks $f_a$ and $f_k$, they are 2-layer MLP with the ReLU activation. The projection network $h$ is a 2-layer MLP with hidden size of 128 and output size of 64, followed by LayerNorm. $\bar{h}$ is also momentum updated from $h$.

#### A.1.2  $\text{CUDC}^{\text{ICM}}_{\text{VARY}}$ and $\text{CUDC}^{\text{APT}}_{\text{VARY}}$

$\text{CUDC}^{\text{ICM}}_{\text{vary}}$ and $\text{CUDC}^{\text{APT}}_{\text{vary}}$ are two variants of the proposed CUDC. Given the implementation of ICM and APT in ExORL (Yarats et al., 2022), we added the forward network $f_s$ and backward network $f_k$ in order to let these models learn more feature representations with the dynamics information. Therefore, the temporal distance of $k$-step can be adapted in a similar way as in CUDC. $k$ is increased by 1 if the proportion of tuples with low intrinsic rewards is greater than a threshold, i.e. $\frac{\sum_i^n \mathbf{1}[r_i < C_r]}{n} > C_k$. In this way, we can investigate the effects of adapting environment step on two unsupervised baseline methods.

#### A.1.3  $\text{CUDC}_{\text{REWARD}}$

$\text{CUDC}_{\text{reward}}$ is a variant of the proposed CUDC. Given the implementation of $\text{CUDC}^{\text{APT}}_{\text{vary}}$ in the previous subsection, we extended the state entropy to state-action entropy and meanwhile add a prediction error of the $k$-step future state as formulated in Equation (3). Therefore, agent can be encouraged to explore more diverse state-action space while focusing on the under states with high prediction errors. In this way, we can investigate the effects of the proposed mixed intrinsic reward.

#### A.1.4  $\text{CUDC}_{\text{REACH}}$

$\text{CUDC}_{\text{reach}}$ is a variant of the proposed CUDC. Given the implementation of $\text{CUDC}_{\text{reward}}$ in the previous subsection, we incorporated the full reachability module with the reachable network. As a result, the adaptive update of the environment step can be facilitated by the curiosity weight $w$ outputted by the reachability module and the enhanced representation learning can be carried out in a self-supervised manner. Compared with the full CUDC model, $\text{CUDC}_{\text{reach}}$ has disabled the regularization of the critic-actor update. In this way, we can investigate the effects of the proposed reachability module.

### A.2  Hyperparameter Setting

#### A.2.1  Data Collection

We provide a full set of common hyperparameters used in baselines as welll as CUDC in Table 2, which closely follows the same settings from ExORL (Yarats et al., 2022) and URLB (Laskin et al., 2021).

For the other hyperparameter used in CUDC, they are listed in Table 3.

It should be noted that Explore2Offline (Lambert et al., 2022) is a concurrent work for data collection, but its source code is not available at the moment.

#### A.2.2  Offline RL

For the offline RL agent, we follow the findings reported in ExORL that even the vanilla off-policy RL algorithm of TD3 (Fujimoto et al., 2018) can outperform the carefully designed offline RL algorithms when the collected dataset is of high quality. Thus, we implement an offline RL of TD3

Table 2: Common hyperparameter setting for the unsupervised data collection methods

| Hyperparameter | Value |
|---|---|
| Observation type | states |
| Replay buffer Size | $10^6$ |
| Action repetitions | 1 |
| Seed frames | 4000 |
| Batch size | 1024 |
| Discount factor | 0.99 |
| Optimizer | Adam |
| Learning rate | $10^{-4}$ |
| Non-linearity | ReLU |
| Agent update frequency | 2 |
| Critic target EMA rate | 0.01 |
| Hidden dimension | 1024 |
| Exploration stddev clip | 0.3 |
| Exploration stddev value | 0.2 |

Table 3: Hyperparameter setting for the proposed CUDC

| Hyperparameter | Value |
|---|---|
| $k$-step range | 3, 4, 5, 6 |
| State representation dimension | 512 |
| Actor representation dimension | 64 |
| MLP hidden dimension for action encoder | 64 for action encoder |
| MLP hidden dimension | 128 for projection |
| Projection dimension | 64 |
| Regularization clip | [0.2, 2] for Walker |
| | [0.2, 1] for Quadruped |
| | [0.9, 1] for Jaco Arm |
| Intrinsic reward weights $(\alpha, \beta)$ | $(10^{-3}, 10^{-2})$ for Walker |
| | $(10^{-4}, 10^{-2})$ for Quadruped |
| | $(100, 0)$ for Jaco Arm |
| K-NN | 12 |
| Threshold $C_w$ | 0.02 for Walker and Quadruped |
| | 0.01 for Jaco Arm |
| Threshold $C_k$ | 512 |

to evaluate the quality of the collected dataset. The detailed implementation and hyperparamter settings can be found in ExORL.

# B    PROOF OF LEMMA 3.1

*Proof.* It has been shown in APT Liu & Abbeel (2021b) that the particle-based entropy estimator of the state representation $z$ can be derived as

$$\mathcal{H}(z) = -\frac{1}{n}\sum_{i=1}^{n}\log\frac{K}{nv_i^K} + b(K) \propto \sum_{i=1}^{n}\log v_i^K \qquad (5)$$

where $b(K)$ represents a bias correlation and $v_i^K$ indicates the volume of the hypersphere with a radius of $||z_i - z_i^{\text{K-NN}}||$ between the $z_i$ and its $K$-th nearest neighbor $z_i^{\text{K-NN}}$.

By substituting $v_i^K = \frac{||z_i - z_i^{\text{K-NN}}||\pi^{n_z/2}}{\Gamma(n_z/2+1)}$ where $\Gamma$ is a gamma function and $n_z$ is the dimension of $z$, we can obtain

$$\mathcal{H}(z) \propto \sum_{i=1}^{n}\log ||z_i - z_i^{\text{K-NN}}||_2. \qquad (6)$$

Let $u = (z_s, z_a)$ represent the state-action representation and we can further substitute $z = u$ into Equation (6) to obtain

$$\mathcal{H}(u) \propto \sum_{i=1}^{n}\log ||u_i - u_i^{\text{K-NN}}||_2. \qquad (7)$$

□

## C  ADDITIONAL RESULTS AND DISCUSSIONS

### C.1  FULL RESULTS OF MAIN EXPERIMENTS

Figure 5 and Figure 6 show the full results on the 12 downstream tasks across 3 domains. Figure 7 summarizes the overall performances in 3 domains. They demonstrate that our proposed methods significantly outperform the baseline methods in 3 downstream tasks of Walker and 3 downstream tasks of Quadruped. However, in the hardest domain of Jaco Arm, the learning sample efficiency becomes poor in 3 downstream tasks although its learning performance at 500K step catches up with the best baseline method of ICM. We explain this by the fact that our method can introduce more complexity and challenges for the agent whenever adapting the temporal distance $k$ to learn a better representation. However, as Jaco Arm is indeed the most challenging domain, the introduced complexity cannot be fully coped with this environment. Thus, the poor performance in sample efficiency can occur, and it is interesting to further study on how to cope with hard domain environments in the future works.

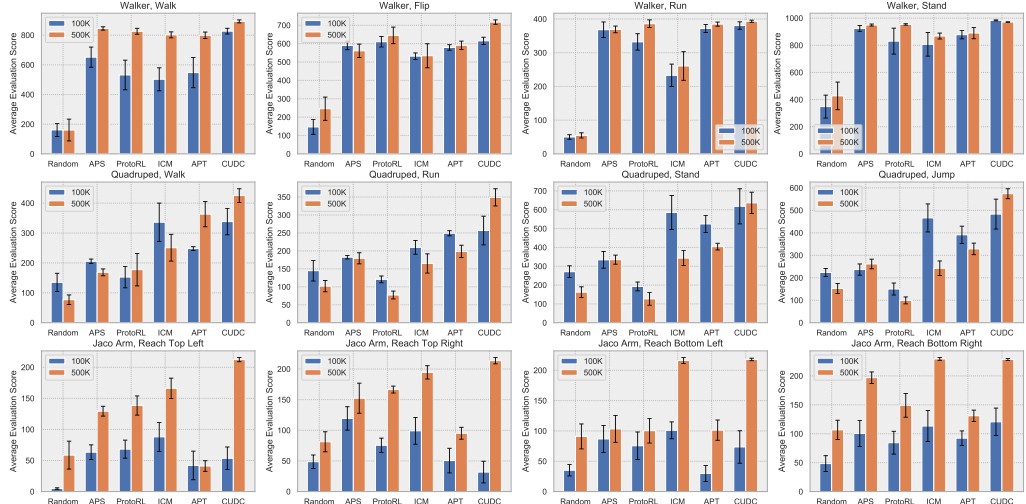

Figure 5: Full results of the average performance score evaluated at 100K and 500K steps for the offline RL agent on each downstream task. CUDC significantly improves the sample-efficiency in 9 out 12 tasks at 100K step and learning performances in all 12 tasks at 500K step.

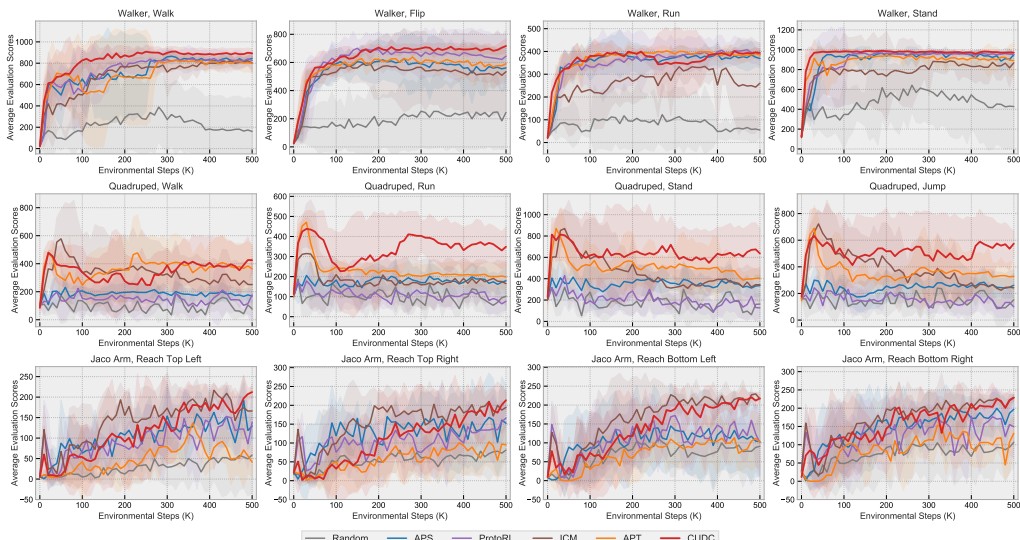

Figure 6: Learning curves of the offline RL agent on full 12 downstream tasks with the task-agnostic dataset collected by different methods. The proposed CUDC demonstrates superior capability of improving the sample-efficiency and learning performances of the offline RL agent in the domains of Walker and Quadruped.

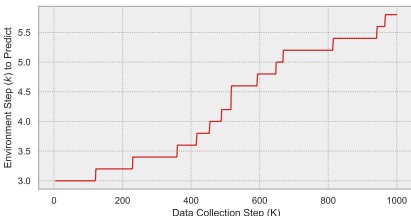

Figure 7: The overall offline learning performance across 3 domains. CUDC consistently leads to significant improvement in offline agent's performance at 500K step in all 3 domains on average.

In Figure 8, we plot how our proposed method adapts the $k$-step during data collection in the Walker domain. It can be observed that agents take around 450K step to learn and adjust the temporal distance from $k = 3$ to $k = 4$. Then, it takes around just 200K steps to increase from $k = 4$ to $k = 5$. Finally, it takes more than 400K steps to reach $k = 6$. This increasing behavior implies that at the early phase of training, we should not inject a too challenging $k$ value. Once the agent has learned enough dynamics information, they are able to learn quickly on more challenging reachability analysis. However, after a certain stage ($k = 5$ in this domain), the learning becomes too difficult for the agent. Therefore, it is interesting to further adapt this over-difficult knowledge in the future works as well.

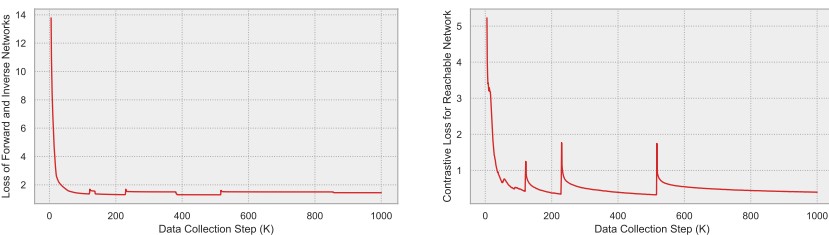

Figure 8: The adaptive increase of how many steps into the future that the dynamics model should predict. It is averaged by 5 random runs in the Walker domain, where the k-step is limited to change from 3 to 6.

In Figure 9, we visualize how the loss of dynamics model (forward and inverse networks) and the contrastive loss of the reachable networks change w.r.t. training steps. Both losses decrease and converge during learning. In particular, they only increase when the step $k$ is adapted to increase. After that, both losses decrease quickly. The agent can well predict each sampled $k$-step future state being most reachable from its own current state rather than those from other transition tuples. Even without inputting the sequence of intermediate actions or states, the agent can predict the $k$-step future state accurately. During the model training, the feature representations are enhanced as well. Therefore, it can be validated that the learned representation contains more semantic latents with dynamics information, with the self-supervised learning of agent's internal belief.

Figure 9: The visualization of the dynamics model loss as well as reachability loss (contrastive loss) during learning in the Walker domain.

In addition to the offline multi-task learning performance, we compare the quality of the collected datasets by plotting the normalized density of the true reward for the downstream task of Stand in Walker environment. The visualization is shown in Figure 10. It can be seen that the dataset collected by our proposed CUDC is of higher quality, with larger density for high rewards and a larger proportion at the low reward part. Specifically, the mean trajectory reward of CUDC is

0.312, which is 69% higher than APS, 21% higher than ProtoRL, 16% higher than ICM, and 10% higher than APT. Moreover, the 75% quartile of the trajectory reward for CUDC is 0.498, which is 143% larger than APS, 42% larger than ProtoRL, 19% larger than ICM, 16% larger than APT. Therefore, it indicates the effectiveness of our proposed method to collect higher-quality dataset, where increasingly more rewarding states have been visited.

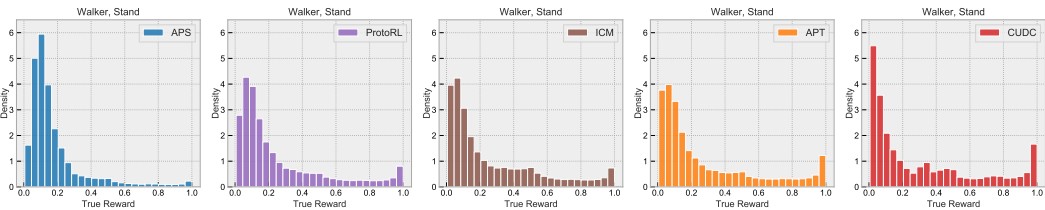

Figure 10: The visualization of the true reward density on Walker, Stand task for the task-agnostic dataset collected by the proposed method and baseline models.

## C.2 EVALUATION BY ANOTHER OFFLINE RL ALGORITHM OF CQL

In this work, we consider the problem setting for offline RL into three main steps: data collection, reward relabeling, and downstream offline learning. Our work focuses only on the first step of collecting high-quality dataset and thereby is agnostic to the offline RL algorithms. For the main experiments, we chose TD3 (Fujimoto et al., 2018) to evaluate the multi-task downstream learning, since it was concluded in ExORL that the vanilla TD3 algorithm can effectively learn offline and even outperform carefully designed offline RL algorithms by improving the dataset quality.

To demonstrate that the dataset collected by our proposed method is of high quality than the other methods, we conduct additional experiments using another offline RL algorithm of CQL (Kumar et al., 2020) that regularizes the Q-values during training. The results are shown in Figure 11 and our proposed CUDC has also demonstrated improved sample efficiency at 100K learning steps and learning capability at 500K learning steps in all 4 downstream tasks against baseline methods. By comparing the performance score at 100K and 500K steps of CUDC against the best baseline method, CUDC has achieved on average 18.2% and 12.3% improvement at respectively 100K and 500K CQL agent learning steps, across 4 downstream tasks. Specifically, CUDC is 21% higher than the best baseline of APT at 100K steps on Run task while its learning performance is 15% higher than the best baseline of APS at 500K steps.

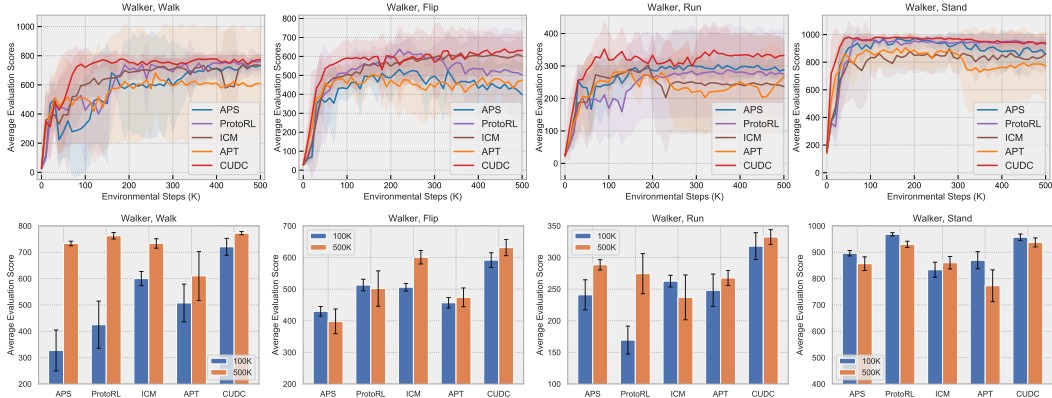

Figure 11: Learning curves (top) and performance scores at 100k and 500K learning steps (bottom) for the CQL agent on 4 downstream tasks of Walker. The proposed CUDC demonstrates superior capability of improving the sample-efficiency and learning performances of the CQL agent in all 4 tasks.

## C.3 RESULTS OF ADDING EACH PROPOSED COMPONENT ON TOP OF EACH OTHER

We present the performance scores of the four variants of CUDC at 100K and 500K steps in Table 4. Moreover, the detailed learning curves of the offline RL agent on 4 downstream tasks are shown in Figure 6. It can be concluded that adapting the temporal distance to reach more distant $k$-step future can substantially improve both sample efficiency and learning capabilities. Moreover, all proposed components facilitated by the proposed reachability module are necessarily important to yield the improvement. As a result, the full CUDC model further addressed the instability issue to obtain the minimum standard deviation among all methods.

Table 4: Performance scores (mean & standard deviation) of four versions of CUDC at 100K and 500K environment steps. The full model outperforms other versions on 3 out of 4 tasks in sample efficiency (100K) and all four tasks in learning performance (500K) regimes, across 5 random seeds.

| 100K Step Score | ICM | CUDC$_{vary}^{ICM}$ | APT | CUDC$_{vary}^{APT}$ | CUDC$_{reward}$ | CUDC$_{reach}$ | CUDC |
|---|---|---|---|---|---|---|---|
| Walker, Walk | 503±258 | 686±46 | 548±338 | 785±56 | 796±85 | 811±69 | **827±64** |
| Walker, Flip | 530±59 | 589±82 | 579±51 | 542±173 | 601±156 | 570±71 | **615±66** |
| Walker, Run | 233±111 | 320±26 | 372±40 | **387±19** | 385±69 | 370±40 | 381±37 |
| Walker, Stand | 797±312 | 909±88 | 878±101 | 936±69 | 950±40 | **983±11** | **984±11** |
| 500K Step Score | ICM | CUDC$_{vary}^{ICM}$ | APT | CUDC$_{vary}^{APT}$ | CUDC$_{reward}$ | CUDC$_{reach}$ | CUDC |
| Walker, Walk | 802±67 | 822±52 | 799±73 | 820±71 | 824±52 | 849±36 | **893±34** |
| Walker, Flip | 534±218 | 665±32 | 591±77 | 674±77 | 666±53 | 679±41 | **717±41** |
| Walker, Run | 261±142 | 336±115 | 384±20 | 370±38 | **394±43** | 375±27 | 393±11 |
| Walker, Stand | 868±73 | 927±74 | 890±139 | 916±78 | **970±12** | 969±11 | **971±7** |

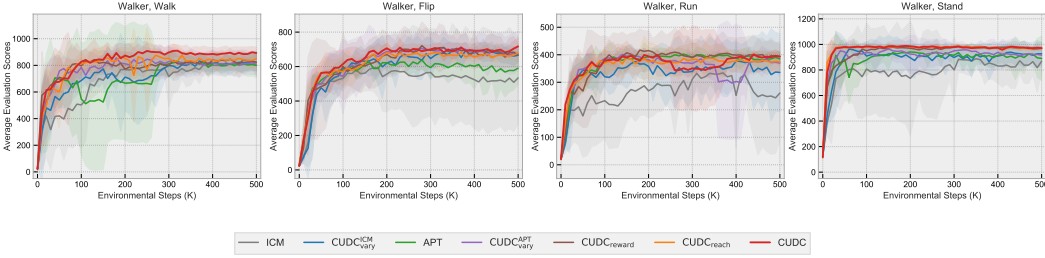

Figure 12: Full results of learning curves on 4 downstream tasks of Walker environment with the task-agnostic dataset collected by different versions of the proposed CUDC. They are capable of improving both sample efficiency at 100K step and learning capabilities at 500K steps, compared to the baselines of ICM and APT. All proposed components work inherently to collect the high quality dataset for offline learning.

## C.4 RESULTS OF REMOVING EACH PROPOSED COMPONENT FROM FULL MODEL

To further investigate the effectiveness of each proposed component and quantify the importance of them, we carry out additional experiments by respectively removing each proposed component from the full model. In particular, the following models are used to collect the task-agnostic dataset for the Walker environment with 5 random seeds.

- CUDC$_{-Entropy}$: The proposed intrinsic reward of KNN-based particle entropy of state and action $r_{\mathcal{H}}(s_t, a_t)$ is removed, and only the prediction error based reward $r_{\mathcal{E}}(s_t, a_t)$ is used.
- CUDC$_{-PE}$: The proposed intrinsic reward of prediction error $r_{\mathcal{E}}(s_t, a_t)$ is removed, and only the KNN-based particle entropy reward of state and action $r_{\mathcal{H}}(s_t, a_t)$ is used.
- CUDC$_{-Regularize}$: the mechanism of regularizing the backbone DDPG algorithm is removed.
- CUDC$_{-Vary}$: $k$-step is fixed to be 3 throughout learning.

- CUDC$_{-\text{Reach}}$: The proposed reachability module is removed while $k$-step is still adapted through the loss of the dynamics model.
- CUDC$_{-\text{Inverse}}$: The inverse networks of predicting the action and step $k$ are removed.

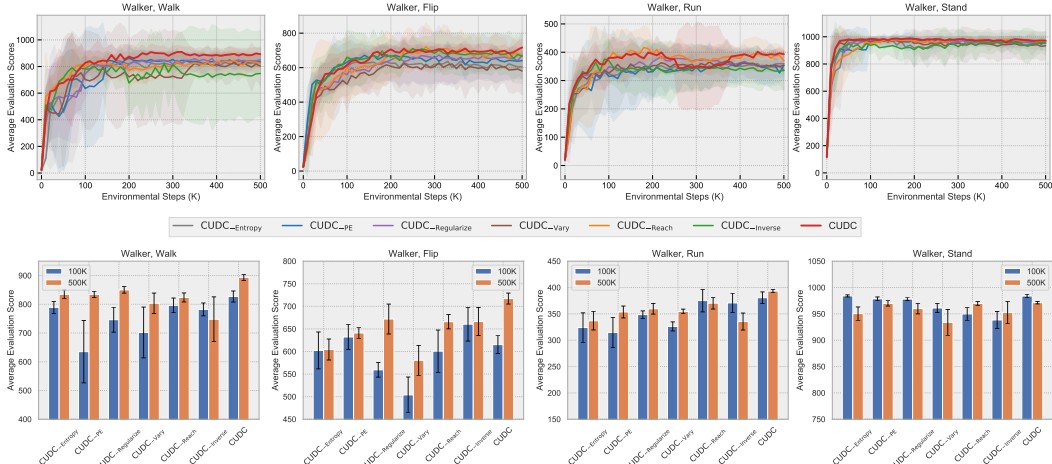

Figure 13: Learning curves (top) and performance scores at 100K and 500K learning steps (bottom) on 4 downstream tasks of Walker environment with the task-agnostic dataset collected by different models of removing each proposed component from CUDC. Removing any proposed component will cause less desirable performance throughout the offline multi-task learning process.

Table 5: Relative performance of the average scores for the ablation models (removing each proposed component from the full CUDC model) at 100K and 500K learning steps. Adapting k-step is the most effective component to the improved sample efficiency and learning capabilities, while the proposed mixed intrinsic reward is the second.

| 100K Relative Performance | CUDC$_{-\text{Entropy}}$ | CUDC$_{-\text{PE}}$ | CUDC$_{-\text{Regularize}}$ | CUDC$_{-\text{Vary}}$ | CUDC$_{-\text{Reach}}$ | CUDC$_{-\text{Inverse}}$ | CUDC |
|---|---|---|---|---|---|---|---|
| Walker, Walk | 0.954 | **0.768** | 0.902 | 0.849 | 0.963 | 0.946 | 1.000 |
| Walker, Flip | 0.979 | 1.028 | 0.909 | **0.819** | 0.976 | 1.074 | 1.000 |
| Walker, Run | 0.851 | **0.827** | 0.916 | 0.857 | 0.986 | 0.974 | 1.000 |
| Walker, Stand | 1.000 | 0.995 | 0.994 | 0.977 | 0.965 | **0.954** | 1.000 |
| Mean | 0.946 | 0.904 | 0.930 | **0.875** | 0.973 | 0.987 | 1.000 |
| 500K Relative Performance | CUDC$_{-\text{Entropy}}$ | CUDC$_{-\text{PE}}$ | CUDC$_{-\text{Regularize}}$ | CUDC$_{-\text{Vary}}$ | CUDC$_{-\text{Reach}}$ | CUDC$_{-\text{Inverse}}$ | CUDC |
| Walker, Walk | 0.935 | 0.934 | 0.952 | 0.900 | 0.922 | **0.837** | 1.000 |
| Walker, Flip | 0.843 | 0.894 | 0.937 | **0.809** | 0.929 | 0.929 | 1.000 |
| Walker, Run | 0.856 | 0.899 | 0.914 | 0.902 | 0.942 | **0.853** | 1.000 |
| Walker, Stand | 0.979 | 0.999 | 0.989 | **0.962** | 0.999 | 0.981 | 1.000 |
| Mean | 0.903 | 0.931 | 0.948 | **0.893** | 0.948 | 0.900 | 1.000 |

Figure 13 shows the overall offline learning performances. We can observe that limiting $k = 3$ and removing the inverse networks will cause the most significant performance decreases among the evaluated models. It implies that adapting the step between current and future states can help to learn rich representation, which plays the most important role in our proposed CUDC to collect high-quality dataset for offline multi-task learning. Meanwhile, the inverse networks are necessary to learn the representation that is robust to the uncontrollable features by the agent's actions and enables the encoders to capture the dynamics information in the learned representation.

To quantitatively measure the benefits brought by each proposed component, we compute the relative performance scores at 100K for sample efficiency and 500K for learning capabilities based on the full model. The results are summarized in Table 5. For the sample efficiency at 100K steps, removing the k-step adaptation has caused the worst performance with a 18.1% decrease in Flip task and an overall 12.5% decrease across all 4 downstream tasks, followed by removing prediction-error-based reward (9.6%), removing regularization (7.0%) and removing entropy-based reward (5.4%).

For the learning capabilities at 500K steps, removing the k-step adaptation has also resulted in the worst performance with a 19.1% decrease in Flip task and an overall 10.7% decrease across all 4 downstream tasks, followed by removing inverse networks (10.0%), removing entropy-based reward (9.7%) and removing prediction-error-based reward (6.9%). Thus, we conclude that adapting how many steps into the future that the dynamics model should predict is most effective to the improved sample efficiency and learning capabilities, while the proposed mixed intrinsic reward is the second most effective.

## C.5 ABLATION STUDIES ON THE RANGE OF VARYING $k$-STEP

In our work, we set the range of varying k-step from 3 to 6. It starts from 3 as we follow the same setting as the ExORL benchmark, where all 8 data collection baselines strictly limit k=3 for the future state in each transition tuple. As for the threshold of $C_w$ and $C_k$, we did not specifically hypertune these values and they were just set to ensure that $k$ can be varied from 3 to 6 adaptively during the 1M dataset collection process. As for the upper bound of 6, it is an optimal end value to obtain the desired performances. To support this finding, we carry out a sweep of the upper bound of $k$ from 3 to 8, and present the results in Figure 14. Firstly, it can be observed that the learning capabilities at 500K learning steps first increase and then decrease with the increase of the upper bound of $k$, across all 4 downstream tasks. Setting the upper bound of 6 achieves the highest in Walk and Run tasks while it is the second highest in Flip and Stand tasks. Secondly, there is no clear trend of performance at 100K steps by setting different values of the upper bound for $k$. It can be explained by the complexity caused by varying the $k$-step for the future states. However, we can still observe that setting the upper bound of 6 achieves the highest in 3 tasks. Therefore, we believe $k$ should vary from 3 to 6 to learn rich representation with more semantic latents.

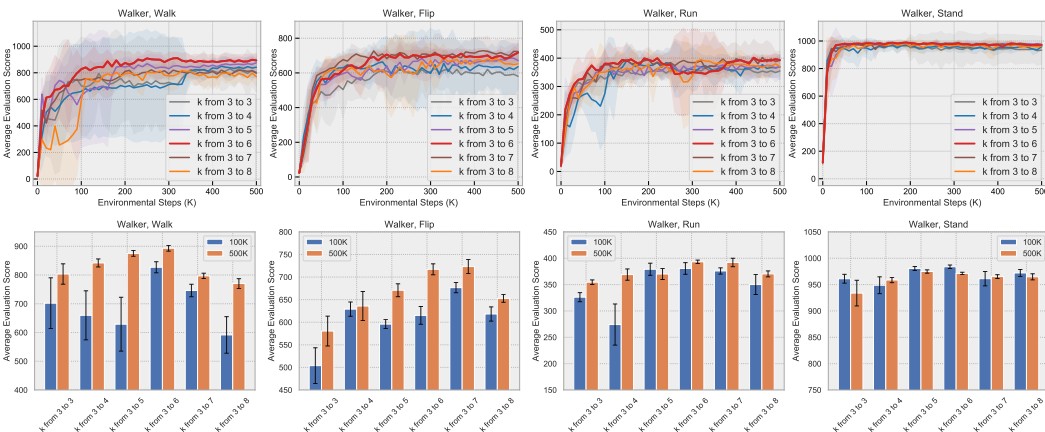

Figure 14: Learning curves (top) and performance scores at 100K and 500K steps (bottom) on 4 downstream tasks of Walker with the task-agnostic dataset collected by different range of $k$-step. Overall, setting the range from 3 to 6 performs relatively the best.

