# OpenReview forum: "Curiosity-Driven Unsupervised Data Collection for Offline Reinforcement Learning"
_ICLR.cc/2023/Conference — Submitted to ICLR 2023_

### Official Review · Reviewer_S2Ud · 2022-10-22

**Confidence:** 3
**Correctness:** 4
**Technical Novelty And Significance:** 3
**Empirical Novelty And Significance:** 3
**Recommendation:** 6

**Clarity, Quality, Novelty And Reproducibility:**

Clarity: The paper is well written and the approach is well explained. Fig. 1 in particular is very helpful to understand the overall approach.

Quality & Novelty: The presented approach is novel in its application of environment step adaptation for generating curiosity based rewards and importance weighting for collecting datasets for offline RL. The approach is well evaluated and the ablations are well done; the results are also promising.

Reproducibility: Details regarding the different network architectures and hypers are provided which can make it easy to reproduce the results but since the approach has several moving parts it can be tricky to reproduce.

**Details Of Ethics Concerns:**

No concerns.

**Strength And Weaknesses:**

Strengths:
1. The approach is well motivated and different parts of the proposed approach are analyzed reasonably well in the ablation experiments. Particularly, the adaptation of the environment step and its application to data-collection for offline RL is quite interesting and seems to aid downstream performance.
2. The evaluation is thorough (albeit on a simple set of tasks) and the learned policies show significant improvements in performance compared to prior work. Ablations clearly demonstrate the importance of the different choices made.

Weaknesses:
1. The approach is a bit complex and has several moving parts. In particular it is not clear if the inverse networks are necessary as no explicit ablation is done removing them, and the intrinsic bonus and importance weights do not directly depend on these predictions.
2. While most results show downstream agent performance after offline RL using TD3, it would have been useful to see some quantification of the (downstream task) reward within the collected datasets. This can be a good way to directly compare how useful the dataset is for each of the downstream tasks; plotting this against data collection progress on the x-axis can also help quantify if the change in environment step leads to increasingly more rewarding states being seen.
3. The adaptation of the environment step is a key novelty in the proposed approach and while it is ablated against different schemes (and different intrinsic rewards) it would be helpful if results are provided for a sweep over hypers for the specific formulation chosen in the paper (specifically, the start/end values of k, and threshold C_w and C_k).
4. The intrinsic reward weights \alpha and \beta for the Jaco task are orders of magnitude different from the others, is there a specific reason for this choice?

**Summary Of The Paper:**

This paper presents a curiosity driven unsupervised data collection method for collecting task-agnostic datasets for offline reinforcement learning. The approach proposes a learnable state reachability module that provides an intrinsic reward for exploration by combining a state-action entropy bonus with a forward dynamics prediction error based reward; the "horizon" or "step" of the forward dynamics prediction is adapted over time to enable the method to focus on more challenging dynamics over time which serves to better condition the learned representations and aid exploration. Additionally, the learned reachability module provides an importance weight based on the estimated reachability; this is used to weight both the actor and critic updates and enables the agent to focus on less-explored parts of the state space. The approach is used together with DDPG for collecting data, the resulting dataset is relabeled and used for offline RL using TD3. The performance of these learned policies are compared with several baselines and ablations on three domains with four tasks each; results show significant improvement compared to baselines.

**Summary Of The Review:**

Overall, this approach presents an interesting application of curiosity driven exploration for data collection for offline RL. The results are promising and the evaluation is well done. I would suggest a weak accept.

---

> ### Author Response · Authors · 2022-11-18
> **Response to Reviewer S2Ud (1/2)**
>
> ### Q1: It is not clear if the inverse networks are necessary
> **A1**: We have added more experimental results in Figure 13 and Table 5 of Appendix C.4 in the updated version, where the inverse networks are removed. As explained in ICM Section 2.2 [a], the inverse network of action helps to learn the representation that is robust to the uncontrollable features by the agent’s actions, although it is not directly used in the intrinsic reward formulation. Similarly, we added the inverse network of k-step prediction, which enables the encoders to capture the dynamics information in the learned representation. Quantitatively, Table 5 shows that the learning capabilities at 500K steps overall decrease by **10%**, across 4 downstream tasks of Walker on the dataset collected by the method without the inverse networks. Specifically, it drops the most by **16.3%** in Walk task and **14.7%** in Run task. Thus, we can conclude that the proposed inverse networks implicitly play an important role in learning richer representation of dynamics information to collect high-quality dataset for offline RL.
>
> [a] Pathak, Deepak, et al. "Curiosity-driven exploration by self-supervised prediction." International conference on machine learning. PMLR, 2017.
>
> ### Q2: Compare how useful the dataset is for each of the downstream tasks
> **A2**: Thank you for your suggestion.
> 1) To demonstrate that the dataset collected by our proposed method is of higher quality than the other methods, we conduct additional experiments using another offline RL algorithm CQL that regularizes the Q-values during training. The new results are shown in Figure 11 of Appendix C.2, where our proposed CUDC has also demonstrated improved sample efficiency at 100K learning steps and superior learning capability at 500K learning steps in all 4 downstream tasks. By comparing the performance scores at 100K and 500K steps of CUDC against the best baseline method, CUDC has achieved on average **18.2% and 12.3% improvement** at respectively 100K and 500K learning steps, across 4 downstream tasks. In particular, our proposed CUDC is **21.1% higher** than the best baseline of APT at 100K steps on Run task while its learning performance is **15.2% higher** than the best baseline of APS at 500K steps. Hence, we conclude that the collected datasets by CUDC allows for better multi-task learning.
>
> 2) We have added the true reward density visualization for the downstream task of Stand with the task-agnostic dataset to compare our proposed method with baseline methods. The normalized density for the dataset collected by different methods is shown in Figure 10 of Appendix C.1 in the updated version. Compared to the baseline methods, the dataset collected by CUDC has larger density for high rewards; meanwhile, it also has a larger proportion at the low reward part. Specifically, the mean trajectory reward of CUDC is 0.312, which is **69% higher** than APS, **21% higher** than ProtoRL, **16% higher** than ICM, and **10% higher** than APT. Moreover, the 75% quartile of the trajectory reward for CUDC is 0.498, which is **143% larger** than APS, **42% larger** than ProtoRL, **19% larger** than ICM, and **16% larger** than APT. Therefore, it indicates the effectiveness of our proposed method to collect higher-quality dataset, where increasingly more rewarding states have been visited.

---

> ### Author Response · Authors · 2022-11-18
> **Response to Reviewer S2Ud (2/2)**
>
> ### Q3: A sweep over hypers for the specific formulation chosen in the paper (specifically, the start/end values of $k$, and threshold $C_w$ and $C_k$).
> **A3**: In our work, we set the range of varying $k$-step from 3 to 6. It starts from 3 as we follow the same setting as the ExORL benchmark, where all 8 data collection baselines strictly limit k=3 for the future state in each transition tuple. As for the threshold of $C_w$ and $C_k$, we did not specifically hypertune these values and they were just set to ensure that $k$ can be varied from 3 to 6 adaptively during the 1M dataset collection process. As for the upper bound of 6, it is an optimal end value to obtain the desired performances. To support this, we carry out a sweep of the upper bound of $k$ from 3 to 8, and present the results in Figure 14 of Appendix C.5 in the updated version. In addition, the following findings can be observed in Figure 14.
>
> 1) The learning capabilities at 500K learning steps first increase and then decrease with the increase of the upper bound of $k$, across all 4 downstream tasks. Setting the upper bound of 6 achieves the highest in Walk and Run tasks while it is the second highest in Flip and Stand tasks.
> 2) There is no clear trend of the performances at 100K steps by setting different values of the upper bound for $k$. It can be explained by the complexity caused by varying the temporal distance of $k$-step to predict the future states. However, we can still observe that setting the upper bound of 6 achieves the highest in 3 tasks.
>
> Therefore, we believe it is optimal to varry $k$ from 3 to 6 to learn rich representation with more semantic latents.
>
> ### Q4: The intrinsic reward weights $\alpha$ and $\beta$ for the Jaco task are orders of magnitude different from the others
> **A4**: Jaco environment is different from the other two locomotion-related environments as it is a 6 degree-of-freedom robotic arm with a three-finger gripper for object manipulation. It is also the most difficult environment among the three. By comparing the performance scores of APT and ICM in Table 1, we can see ICM performs significantly better than APT in Jaco while it is relatively the same or slightly worse than APT in the other two environments. Thus, we set the weights of $\alpha$ and $\beta$ orders of magnitude different in Jaco to encourage agents to focus more on under-learned states with high prediction errors (similar to ICM).

---

### Official Review · Reviewer_LA4m · 2022-10-24

**Confidence:** 3
**Clarity, Quality, Novelty And Reproducibility:** Clarity needs improvement, see above.
**Correctness:** 3
**Technical Novelty And Significance:** 2
**Empirical Novelty And Significance:** 2
**Recommendation:** 5

**Strength And Weaknesses:**

*Strengths*
- How to design unsupervised exploration/data collection methods to facilitate offline RL on that data is an important problem.
- The proposed method does seem to produce better datasets for offline RL on Mujoco tasks than the baselines.

*Weaknesses*

Clarity/Method: The main flaw in the paper is the clarity of presentation. The paper proposes a complicated method with many moving parts but fails to motivate or clearly explain them.
- A large part of the introduction and abstract proposes the "k-step" exploration as the key advancement proposed in this work. However it is not clear from much of the paper what exactly is meant by this (e.g. action repeats for k steps, planning length k sequences at a time, extending the horizon of each episode, etc.) Ultimately, it seems like the only impact the choice of k has is in choosing how many steps into the future the dynamics model should predict. The paper could do a much better job of clearly explaining this from the beginning, and probably should choose some different term than "environment step".
- Its also not very convincing that the scheme for adapting this environment step is necessary or effective. The paper proposes to start the value of k at 3 and gradually increase it to 6, with some thresholds on the intrinsic reward to determine when to increase it. This feels a bit handcrafted, and looking at FIgure 4 it doesn't seem convincingly better than the simple uniform or gaussian baseline.
- The dynamics model takes as input a sequence of k actions correct? The notation seems to suggest its just one action, but predicting s_{t+k} seems ill posed without a sequence of k actions.
- Overall the method seems to be taking 3 established types of exploration objectives (reachability, entropy, and model error), and combining them by summing the entropy and model error rewards, then scaling the Q function loss by the reachability reward. This way of combining things feels a bit arbitrary. Why is this the right way to combine these parts of the objective? And why should all these be combined, it seems there could be redundancy between then? The paper should better motivate these design choices.
- Figure 1 is very difficult to parse.

Experiments:
- Are there error bars for figure 3?
- I additionally found the ablation naming quite confusing. I would suggest renaming them based on what exactly is being removed from the method in each.
- Overall the experiments seem to show that the proposed method outperforms the baselines and ablations, though I'm not 100% sold on these experiments due to (1) the ablations seem incomplete (don't include removing each of the 3 parts of the exploration reward) and (2) the minimal gains/no error bars in the ablation experiments.

**Summary Of The Paper:**

This paper studies the problem of unsupervised exploration in RL, such that the resulting dataset is good for downstream learning (specifically offline RL). The majority of the technical contribution of the paper is in a method for unsupervised exploration.

The proposed method has quite a few moving parts.
- First, it learns a state and action representation with a contrastive dynamics modeling loss, where conditioned on timesteps k, a model predicts the state reached k steps into the future after applying k actions. It trains such that the predicted state is closer to the embedding of the true s_{t+k} than negatives.
- It also trains inverse dynamics models and temporal distance predictors from these low-dimensional embeddings.
- It simultaneously estimates the particle-based entropy of the state-action distribution using K nearest neighbors distance.
- The intrinsic exploration reward is formulated as a weighted sum of the particle-based entropy, and the prediction error of the dynamics models (forward, inverse, temporal distance)
- Finally, when training DDPG with intrinsic reward, during policy optimisation the Q values are weighted inversely proportionally to the "reachability" as measured by the k-step dynamics model, up weighting the reward on less reachable states.
The method also has a curriculum which updates the choice of k over the course of training to be further away.

In experiments, they run the proposed approach and other ablations and baseline exploration algorithms for 1M steps, then train offline RL agents on the collected data and evaluate their performance. The proposed approach appears to improve over methods like ICM and APT.

**Summary Of The Review:**

Overall, the paper proposes an interesting exploration algorithm. However the method has many moving parts that are not clearly explained or motivated, and overall the presentation of the paper and clarity of the writing needs significant improvement. The method itself seems to work decently, but its a bit hard to evaluate the experiments/ablations with the current presentation.

---

> ### Author Response · Authors · 2022-11-18
> **Response to Reviewer LA4m (1/2)**
>
> ### Q1: Clarity of $k$-step
> **A1**: In our work, we define the "$k$-step" exploration as to predict the $k$-th distant future state from the current state when updating the internal belief of the proposed reachability module. Note that all existing works in the ExORL benchmark implementation have fixed $k=3$ between current state $s_t$ and future state $s_{t+k}$ to train their models (dynamics model, ensemble of predictive models, contrastive model, etc.), limiting the feature space implicitly. Different from this, our main contribution is to adaptively vary this $k$-step during learning and thereby learn a rich representation with more semantic latents. Many thanks for your suggestion. We have included more clarification of $k$-step in the abstract as well as the last three paragraphs of Section 1 for the updated version. In addition, we have changed the expression of “environment steps” to “temporal distance between current and next states” to avoid confusion.
>
>
> ### Q2: It's also not very convincing that the scheme for adapting this environment step is necessary or effective.
> **A2**: Note that the gaussian baseline (Normal) is another version of our method to adapt the value of k. We initialize the mean of Normal to be 3 and sample the value of $k$ from $\mathcal{N}(3,1)$. When the adapting scheme proposed in Section 3.4.1 is met, we increase the mean by 1 and keep sampling the value of $k$ from e.g. $\mathcal{N}(4,1)$. Therefore, it should be considered as an alternative adapting scheme of our proposed method. As shown in Figure 4, Normal even outperforms our proposed method in the Walker_Flip task. However, its overall performance across all 4 downstream tasks is still **4.5% weaker** than ours and its standard deviation is **128% higher** than CUDC, indicating the instability issue caused by this sampling method.
>
> In contrast, the baseline of Uniform does not contain any adaptation scheme and it performs much worse than our proposed CUDC. Figure 4 shows that Uniform only achieves **81%** of sample efficiency as CUDC at 100K steps and **85%** performance scores as CUDC at 500K steps, across 4 downstream tasks in Walker. As for the learning stability, Uniform has around **200%** and **300%** of standard deviation as CUDC at respectively 100K and 500K steps.
>
> Based on these results, we strongly believe the proposed scheme for adapting the temporal distance of $k$-step is indeed necessary and effective.
>
>
>
> ### Q3: The dynamics model takes as input a sequence of $k$ actions correct?
> **A3**: No, our proposed method does not take a sequence of $k$ actions in the dynamics model. As shown in Figure 1, the forward dynamic network defined in Section 3.3 takes the input of $s_t$, $a_t$, and $k$, rather than the sequence of $k$ intermediate actions to predict the $k$-step future state. It should be noted that in the experimental protocol of the ExORL benchmark, all 8 data collection baselines strictly limit $k=3$ for the future state in each transition tuple. In other words, whenever a transition tuple is sampled in these benchmark methods, only ${s_t, a_t, s_{t+3}}$ will be utilized to train the model (i.e. predict the future state $s_{t+3}$ in ICM, Disagreement and APT), and no intermediate action sequence ($a_{t+1}$ and $a_{t+2}$) or future state sequence ($s_{t+1}$ and $s_{t+2}$) is required to obtain the benchmark performances.
>
> In Section 3.4.1 of the submitted version, we also compare with some recent works such as SPR [a] and SGI [b] that predict multiple steps into the future using the sequence of actions and states. However, this approach requires iterative predictions by calling the forward dynamic network for $k$ times. Different from this, our proposed CUDC can automatically adapt the $k$-step and directly perform the $k$-step future state prediction, without increasing the computational complexity substantially.  As shown in Figure 9 of Appendix C.1 in the updated version, even without inputting the sequence of intermediate actions or states, the agent can predict the $k$-step future state accurately. The loss of the dynamics model keeps decreasing within the same step value. During the model training, the feature representations are enhanced as well. Therefore, it can be validated that our dynamics model is effective, which can be used to learn rich representation with more semantic latents of dynamics information.
>
> [a] Schwarzer, Max, et al. "Data-Efficient Reinforcement Learning with Self-Predictive Representations." International Conference on Learning Representations. 2020.
>
> [b] Schwarzer, Max, et al. "Pretraining representations for data-efficient reinforcement learning." Advances in Neural Information Processing Systems 34 (2021): 12686-12699.

---

> ### Author Response · Authors · 2022-11-18
> **Response to Reviewer LA4m (2/2)**
>
> ### Q4: Why is this the right way to combine these parts of the objective? And why should all these be combined, it seems there could be redundancy between them?
> **A4**: In the paragraph of “Effects of Adapting the $k$-step” and “Effectiveness of the Other Proposed Components” of Section 4 in the submitted version, we gradually added the proposed component (adapting the $k$-step future state, the mixed reward, reachability module, and Q-value regularization) on top of each other to investigate their effects on the overall performance. By comparing the improvement of performance scores and the standard deviation of each model with their corresponding baselines in Table 4, all proposed components are necessary and work coherently to yield the improvement. In particular, varying the $k$-step distant future states during training plays the most significant role of improving both sample efficiency (with **18.3% higher** performance scores at 100K) and learning capability (with **9.3% higher** performance scores at 500K). Meanwhile, the proposed reachability module is the most effective component of stabilizing offline learning, with respectively **47.3%** and **24.7%** standard deviation decreases at 100K and 500K learning steps, as shown in Table 4.
>
> In addition, we conduct additional experiments to investigate the effects of each proposed method in Appendix C.4 of the updated version. By respectively removing each proposed component from the full model, we can conclude that adapting $k$-step is the most effective component to the improved sample efficiency and learning capabilities, while the proposed mixed intrinsic reward is the second.  The new results are shown in Figure 13 and Table 5. To quantitatively measure the benefits brought by each proposed component, we compute the relative performance scores at 100K for sample efficiency and 500K for learning capabilities based on the full model. The full comparison results are summarized in Table 5 with the following main highlights.
>
> 1) For the sample efficiency at 100K steps, removing the $k$-step adaptation has caused the worst performance with a **18.1% decrease** in Flip task and an overall **12.5% decrease** across all 4 downstream tasks, followed by removing prediction-error-based reward (9.6%), removing regularization (7.0%) and removing entropy-based reward (5.4%).
> 2) For the learning capabilities at 500K steps, removing the $k$-step adaptation has also resulted in the worst performance with a **19.1% decrease** in Flip task and an overall **10.7% decrease** across all 4 downstream tasks, followed by removing inverse networks (10.0%), removing entropy-based reward (9.7%) and removing prediction-error-based reward (6.9%).
>
> ### Q5: Figure 1 is very difficult to parse.
> **A5**: Figure 1 is the workflow of our proposed method. The diagram on the left illustrates the overall framework, where a batch of transition tuples is sampled from the accumulated dataset, and then the reachability between the $k$-step future and current states is measured by agent’s internal belief. The reachability module outputs the mixed intrinsic reward to encourage diverse exploration. Meanwhile, it also outputs a curiosity weight to adapt the $k$-step and regularize the backbone RL. The diagram on the right shows how the agent measures and updates its internal belief on the probability of the $k$-step future states being reachable from the current states.
>
> ### Q6: Are there error bars for figure 3?
> **A6**: Yes, we have included the error bars for Figure 3 as well as all bar plots in the updated version. It further demonstrates that our proposed method is superior than other baseline methods with more stable performances.
>
> ### Q7: The ablation naming quite confusing
> **A7**: Thanks for your suggestion. The ablation experiments conducted in the submitted version is to add each proposed component on top of each other. For example, $\text{CUDC}^{\text{ICM}}_{\text{vary}}$ is to vary the $k$-step adaptively according to the original intrinsic reward, whose baseline is the original ICM method. We have defined these models in the paragraph of “Baseline Models” in Section 4. For the additional ablation experiments of removing each proposed component from the full model, we name them following your suggestion. For example, $\text{CUDC}_\text{{-Regularize}}$ means the regularization component is removed from the full model. The detailed definition of all models and their results are present in Appendix C.4 in the updated version.

---

### Official Review · Reviewer_F8ph · 2022-10-25

**Confidence:** 4
**Clarity, Quality, Novelty And Reproducibility:** The paper is clearly written.
**Correctness:** 3
**Technical Novelty And Significance:** 3
**Empirical Novelty And Significance:** 3
**Recommendation:** 6

**Strength And Weaknesses:**

Strength:
1. The paper tackles an important problem of data collection in the offline RL setting.
2. The proposed self-supervised algorithm is novel and demonstrates good downstream performance.
3. The paper is clearly written.
4. The experiments conducted are solid.

Weakness:
1. My major concern is about how different offline-RL algorithms can benefit from data collection. The paper uses TD3 as the evaluation protocol algorithm. However, I think evaluating at least some more offline RL algorithms and comparing their performance will make the paper more convincing.
2. Another concern is that I feel the ablation in Table. 4 is not super conclusive. I especially feel the performance difference for different ablations is small. I am not sure what conclusion we can draw from the table, about the effectiveness of each component.
3. I am wondering why the authors don't report any results in some D4RL benchmarks (e.g., some mujoco tasks, ant maze tasks or robotics tasks).
4. The Mixed Intrinsic Reward module is not super novel, given that the entropy and inverse dynamics have been well-known.

**Summary Of The Paper:**

The paper tackles the problem of dataset collection in offline RL, while most of the prior work focuses on building more efficient algorithms given a fixed dataset. To be specific, the paper proposes curiosity-driven unsupervised data collection, that helps the agent to collect data in a self-supervised fashion. The paper proposes to collect task-agnostic data independently, then label the dataset with task-specific rewards. The paper conducts experiments on the DeepMind control suite including 12 different downstream tasks, and the proposed method achieves good performance.

**Summary Of The Review:**

Please see the strengths and weaknesses above. I am overall positive about this paper. But I hope the authors can address the points I raised above.

---

> ### Author Response · Authors · 2022-11-18
> **Response to Reviewer F8ph (1/2)**
>
> ### Q1: How different offline-RL algorithms can benefit from data collection.
> **A1**:
> It has been empirically shown in Section 4.2 of ExORL benchmark that TD3 is an effective evaluator for data collection, which consistently outperforms more sophisticated offline RL algorithms (BC, TD3+BC, CRR, and CQL). Thus, we chose TD3 to evaluate the multi-task downstream learning in the submitted version. To further verify the generalization of the conclusion, we conduct additional experiments using another offline RL algorithm CQL. The new results are shown in Figure 11 of Appendix C.2, where our proposed CUDC has also demonstrated improved sample efficiency at 100K learning steps and learning capability at 500K learning steps in all 4 downstream tasks. By comparing the performance scores at 100K and 500K steps of CUDC against the best baseline method, CUDC has achieved on average **18.2% and 12.3% improvement** at respectively 100K and 500K learning steps, across 4 downstream tasks. In particular, our proposed CUDC is **21.1% higher** than the best baseline of APT at 100K steps on Run task while its learning performance is **15.2% higher** than the best baseline of APS at 500K steps. These results are consistent with our main experiments using TD3 as the evaluator. Hence, we can conclude that our collected datasets are of higher quality, which can be used to allow for better multi-task learning.
>
> ### Q2: Table. 4 is not super conclusive.
> **A2**: Table 4 refers to the results by gradually adding the proposed component (adapting the k-step future state, the mixed reward, reachability module, and Q-value regularization) on top of each other. By comparing the improvement of performance scores and the standard deviation of each model with their corresponding baselines in Table 4, varying the k-step distant future states during training plays the most significant role of improving both sample efficiency (with **18.3% higher** performance score at 100K) and learning capability (with **9.3% higher** performance score at 500K). Meanwhile, the proposed reachability module is the most effective component of stabilizing offline learning, with respectively **47.3%** and **24.7%** standard deviation decreases at 100K and 500K learning steps, as shown in Table 4.
>
> In addition, we conduct additional experiments to investigate the effects of each proposed method in Appendix C.4 of the updated version. By respectively removing each proposed component from the full model, we can conclude that adapting k-step is the most effective component to the improved sample efficiency and learning capabilities, while the proposed mixed intrinsic reward is the second.  The new results are shown in Figure 13 and Table 5. To quantitatively measure the benefits brought by each proposed component, we compute the relative performance scores at 100K for sample efficiency and 500K for learning capabilities based on the full model. The full comparison results are summarized in Table 5 with the following highlights.
>
> 1) For the sample efficiency at 100K steps, removing the k-step adaptation has caused the worst performance with a **18.1% decrease** in Flip task and an overall **12.5% decrease** across all 4 downstream tasks, followed by removing prediction-error-based reward (9.6%), removing regularization (7.0%) and removing entropy-based reward (5.4%).
> 2) For the learning capabilities at 500K steps, removing the k-step adaptation has also resulted in the worst performance with a **19.1% decrease** in Flip task and an overall **10.7% decrease** across all 4 downstream tasks, followed by removing inverse networks (10.0%), removing entropy-based reward (9.7%) and removing prediction-error-based reward (6.9%).

---

> ### Author Response · Authors · 2022-11-18
> **Response to Reviewer F8ph (2/2)**
>
> ### Q3: Lack of experiments in D4RL benchmarks (e.g., some mujoco tasks, ant maze tasks or robotics tasks).
> **A3**: As mentioned in Section 3.1, we consider the problem setting for offline RL into three main steps: 1) data collection, 2) reward relabeling, and 3) downstream offline learning. Our work focuses on Step 1 to collect high-quality dataset, while D4RL benchmarks focus on Step3 with Step 1 and 2 being fixed by default. Specifically, D4RL uses previously collected datasets to evaluate different offline RL algorithms. Note that most of the D4RL environments do not support multi-task learning under our problem setting. For example, the dataset collected in a single ant maze task environment cannot be directly used to learn in another ant maze task due to different state space. Our work does not focus on evaluating different offline RL algorithms that the D4RL benchmarks were designed for, but only improves the data collection process. For the diversity of the evaluation environments, the experiments were carried out on 2 locomotion-related domains and 1 robotic manipulation domain, with increasing difficulty.
>
>
> ### Q4: The Mixed Intrinsic Reward is not novel.
> **A4**: Our main contribution is to introduce an adaptive reachability-based method to improve the data collection process in offline RL settings. We enhance representation learning by exploring more distant future states through the k-step adaptation rather than relying on intrinsic rewards. It can be observed from Table 4 and Figure 3 that adapting the k-step future state alone on ICM and APT can respectively lead to richer feature space with improved sample efficiency and learning performances. This benefit is shown to be stronger than that of the mixed intrinsic reward, summarized in Table 5 of Appendix C.4 in the updated version.
>
> The motivation of our proposed intrinsic reward is to encourage exploration on diverse state and action spaces while focusing more on under learned states with high prediction errors. We have conducted additional experiments and presented more results in Figure 13 of Appendix C.4. It shows that using either type of intrinsic reward alone is not sufficient to obtain the desired performance. Specifically, using only entropy-based reward will obtain 90.4% and 93.1% of the performance scores as the full model at 100K and 500K steps, as shown in Table 5. Meanwhile, using only prediction-error-based reward will obtain 94.6% and 90.3% of the performance scores as the full model at 100K and 500K steps. These new results can demonstrate the effectiveness of our proposed intrinsic reward. For more insights of formulating other intrinsic rewards, it is beyond the scope of our paper.

---

### Official Review · Reviewer_bF1X · 2022-10-26

**Confidence:** 4
**Correctness:** 3
**Technical Novelty And Significance:** 2
**Empirical Novelty And Significance:** 2
**Recommendation:** 3

**Clarity, Quality, Novelty And Reproducibility:**

Clarity & Quality

1. The contributions of the paper are explained clearly, including a clear algorithm section that describes all the pieces of the overall method. The paper provides sufficient background material, and is easy to follow. The technical portion seems to be fine (though some choices are not justified, as elaborated on above).

Originality

1. As described above, the intrinsic reward used essentially combines two common approaches in prior work (KNN-state entropy and prediction error). Using reachability of states to reweight transitions for offline Q-learning seems to be different, but if this is the main contribution it hasn't been analyzed using comparison to other offline RL methods / other approaches for estimating reachability (eg - using model disagreement ).

**Strength And Weaknesses:**


Weaknesses

1. Reachability objective justification - The paper motivates its approach by emphasizing the importance of reachability of future states for offline RL, and proceeds to provide an estimate for this using a contrastive loss where predicted features and corresponding ground truth features are positive pairs, with negatives randomly sampled. There is no explanation for this particular choice beyond the intuition that 'each future state should be most reachable from its own current state'. If there is a lot of overlap of states across trajectories, this statement will likely not hold. Further, even if the states are different, they might need to have similar representations for control (eg: objects might be in similar positions).

2. Model disagreement missing - The paper uses ICM as the comparison for knowledge-based exploration. However, ensemble disagreement of predictive models has shown better results [1,2,3]. This approach is prospective as opposed to retrospective, since it does not rely on prediction error but rather on the variance of the prediction across models, and can solve some problems that prediction error based methods struggle on like the noisy-TV problem. The paper does not compare to this, and also does not discuss why they use prediction error in the intrinsic reward term instead of this. Furthermore, related to 1), an ensemble of predictive models provides an estimate of reachability of states (states on trajectories with lower disagreement are likely to be more reachable).

3. Familiar Intrinsic reward formulation - The paper defines the intrinsic reward to be the sum of terms introduced in previous work. The KNN-based particle entropy of state [4] is modified to include state and action representations, but maximizing entropy of action for exploration has also been previously explored [5]. The other part of the intrinsic reward is just the prediction error of the predicted state feature [6]. These have been studied and examined independently in the past, and the paper does not contribute any new insight into how agents should define rewards for themselves.

4. Experiments/Contribution questions - The paper does include extensive comparisons to a set of prior approaches. However what part of the proposed method leads to the stated benefit? Is it mostly because of the combination of prediction error and particle estimation? Or is it because of the reachability estimate used in offline learning? If it's the latter, how does using better offline RL algorithms such as AWAC/IQL compare (since they also reweight transitions)? If the main contribution is a reweighting scheme for offline RL, then can that be studied independently of the self-supervised exploration setting?

Strengths

1. The paper considers an important problem - that of agents that can direct themselves to collect rich data. Agents that can autonomously act in this manner can keep collecting large datasets, since the bottleneck of task/reward specification is removed.




[1] - Pathak, Deepak, Dhiraj Gandhi, and Abhinav Gupta. "Self-supervised exploration via disagreement.
[2] - Sekar, Ramanan, et al. "Planning to explore via self-supervised world models."
[3] - Mendonca, Russell, et al. "Discovering and achieving goals via world models."
[4] Liu, Hao, and Pieter Abbeel. "Behavior from the void: Unsupervised active pre-training.
[5] Haarnoja, Tuomas, et al. "Soft actor-critic: Off-policy maximum entropy deep reinforcement learning with a stochastic actor."
[6] Pathak, Deepak, et al. "Curiosity-driven exploration by self-supervised prediction."


**Summary Of The Paper:**

The paper proposes an approach for self-supervised learning for agents to collect data, which is then relabelled with task reward and used for downstream tasks with offline RL. The approach estimates reachability of states using a contrastive loss where predicted state features and corresponding ground truth features comprise positive pairs (negatives are randomly sampled from the batch). Transitions with higher reachability estimates are prioritized during offline RL training. The intrinsic reward consists of particle-based kNN entropy and prediction error. The paper includes experiments on Walker, Quadruped and Jaco arm-reaching.

**Summary Of The Review:**

While this paper seeks to tackle an important problem, there are key issues that prevent me from recommending acceptance, including justification of the reachability metric, missing discussion/comparison of model disagreement (for both intrinsic reward and reachability), missing analysis of alternate schemes for offline RL and how they compare to the reweighting approach proposed (given the familiar intrinsic reward that combines known terms).

---

> ### Author Response · Authors · 2022-11-18
> **Response to Reviewer bF1X (1/3)**
>
> ### Q1: Reachability objective justification
> **A1**: In Section 2, we have included the related works of quantifying and exploiting reachability in RL. There are indeed many possible ways of modeling reachability; however, these methods can be inefficient with high computational cost, w.r.t. pairwise similarity computations or kernel density estimation. In contrast, leveraging the self-supervised method of contrastive loss to quantify the reachability is simple and efficient. It has been recently introduced in RL [d,e] and is capable of learning rich representation with more semantic latents. In our work, we define the negative samples similar to the approaches of [d,e] such that each sample should be most reachable to its own future state compared to other samples. This intuition of contrastive learning is elaborated in Section 3.2 in the updated version.
>
> For the two corner cases raised by the reviewer, they did not appear in current self-supervised approaches including [d] and [e]. As for our work, 1) it is not likely to obtain a lot of overlap of states across trajectories as our proposed intrinsic reward consists of the KNN-based particle entropy of state and action, avoiding repeatedly exploring the same state. 2) The learned representation contains more semantic latents and no such similar representation is empirically found (which may confuse the agent during learning). To validate this, we visualize the proposed contrastive loss during learning in Figure 9 (right) of Appendix C.1. This loss keeps decreasing within the same k-step, indicating the agent can well predict each sampled k-step future state being most reachable from its own current state rather than those from other transition tuples.
>
>
> ### Q2: Why not compared to model disagreement?
> **A2**: We agree that using variance of predictive models as an intrinsic reward is particularly vital in many RL problems such as sparse rewards and noisy TV issues. However, it has been empirically shown in the ExORL benchmark [c] that Disagreement [a] performs much worse than ICM when collecting a task-agnostic dataset for multi-task offline RL under our problem setting. Specifically, the results are presented in Figure 2, 8, 9, and 10 in [c]. Figure 2 and 8 even indicate that Disagreement has failed in the goal reaching environment with close-to-zero scores in all 4 downstream tasks and all 5 offline RL algorithms. For this reason, we decide not to include model disagreement as the baseline since ICM has shown to be more desirable among the knowledge-based methods.
>
> ### Q3: Similar intrinsic reward formulation
> **A3**: Our main contribution is to introduce an adaptive reachability-based method to improve the data collection process in offline RL settings. We enhance representation learning by exploring more distant future states through the k-step adaptation rather than relying on intrinsic rewards. It can be observed from Table 4 and Figure 3 that adapting the k-step future state alone on ICM and APT can respectively lead to richer feature space with improved sample efficiency and learning performances. This benefit is shown to be stronger than that of the mixed intrinsic reward, summarized in Table 5 of Appendix C.4 in the updated version.
>
> The motivation of our proposed intrinsic reward is to encourage exploration on diverse state and action spaces while focusing more on under learned states with high prediction errors. We have conducted additional experiments and presented more results in Figure 13 of Appendix C.4. It shows that using either type of intrinsic reward alone is not sufficient to obtain the desired performance. Specifically, using only entropy-based reward will obtain 90.4% and 93.1% of the performance scores as the full model at 100K and 500K steps, as shown in Table 5. Meanwhile, using only prediction-error-based reward will obtain 94.6% and 90.3% of the performance scores as the full model at 100K and 500K steps. These new results can demonstrate the effectiveness of our proposed intrinsic reward. For more insights of formulating other intrinsic rewards, it is beyond the scope of our paper.
>
> [a] - Pathak, Deepak, Dhiraj Gandhi, and Abhinav Gupta. "Self-supervised exploration via disagreement.
>
> [b] Kumar, Aviral, et al. "Conservative q-learning for offline reinforcement learning." Advances in Neural Information Processing Systems 33 (2020): 1179-1191.
>
> [c] Denis Yarats, David Brandfonbrener, Hao Liu, Michael Laskin, Pieter Abbeel, Alessandro Lazaric, and Lerrel Pinto. Don’t change the algorithm, change the data: Exploratory data for offline reinforcement learning. arXiv preprint arXiv:2201.13425, 2022.
>
> [d] Liu, Hao, and Pieter Abbeel. "Behavior from the void: Unsupervised active pre-training.
>
> [e] Srinivas, Aravind, Michael Laskin, and Pieter Abbeel. "Curl: Contrastive unsupervised representations for reinforcement learning." arXiv preprint arXiv:2004.04136 (2020).

---

> ### Author Response · Authors · 2022-11-18
> **Response to Reviewer bF1X (2/3)**
>
> ### Q4.1: What part of the proposed method leads to the stated benefit?
> **A4.1**: In the paragraph of “Effects of Adapting the k-Step” and “Effectiveness of the Other Proposed Components” of Section 4 in the submitted version, we gradually added the proposed component (adapting the k-step future state, the mixed reward, reachability module, and Q-value regularization) on top of each other to investigate their effects on the overall performance. By comparing the improvement of performance scores and the standard deviation of each model with their corresponding baselines in Table 4, all proposed components are necessary and work coherently to yield the improvement. In particular, varying the k-step distant future states during training plays the most significant role of improving both sample efficiency (with **18.3% higher** performance scores at 100K) and learning capability (with **9.3% higher** performance scores at 500K). Meanwhile, the proposed reachability module is the most effective component of stabilizing offline learning, with respectively **47.3%** and **24.7%** standard deviation decreases at 100K and 500K learning steps, as shown in Table 4.
>
> In addition, we conduct additional experiments to investigate the effects of each proposed method in Appendix C.4 of the updated version. By respectively removing each proposed component from the full model, we can conclude that adapting k-step is the most effective component to the improved sample efficiency and learning capabilities, while the proposed mixed intrinsic reward is the second.  The new results are shown in Figure 13 and Table 5. To quantitatively measure the benefits brought by each proposed component, we compute the relative performance scores at 100K for sample efficiency and 500K for learning capabilities based on the full model. The full comparison results are summarized in Table 5 with the following main highlights.
>
> 1) For the sample efficiency at 100K steps, removing the k-step adaptation has caused the worst performance with a **18.1% decrease** in Flip task and an overall **12.5% decrease** across all 4 downstream tasks, followed by removing prediction-error-based reward (9.6%), removing regularization (7.0%) and removing entropy-based reward (5.4%).
> 2) For the learning capabilities at 500K steps, removing the k-step adaptation has also resulted in the worst performance with a **19.1% decrease** in Flip task and an overall **10.7% decrease** across all 4 downstream tasks, followed by removing inverse networks (10.0%), removing entropy-based reward (9.7%) and removing prediction-error-based reward (6.9%).
>
> ### Q4.2: Why not use other offline RL algorithms such as AWAC/IQL compare (since they also reweight transitions)?
> **A4.2**: As mentioned in Section 3.1, we consider the problem setting for offline RL into three main steps: data collection, reward relabeling, and downstream offline learning. Our work focuses only on the first step of collecting high-quality dataset and thereby is agnostic to the offline RL algorithms. We chose TD3 to evaluate the multi-task downstream learning in the submitted version, since it was concluded in ExORL Section 4.2 that the vanilla TD3 can effectively learn offline and even outperform carefully designed offline RL algorithms by improving the dataset quality [c].
>
> To further verify the generalization of our method, we conduct additional experiments using another offline RL algorithm CQL [b] that regularizes the Q-values during training. Note that AWAC and IQL can be used as benchmarks as well but unfortunately their source codes are not available at the moment. The new results are shown in Figure 11 of Appendix C.2, where our proposed CUDC has also demonstrated improved sample efficiency at 100K learning steps and learning capability at 500K learning steps in all 4 downstream tasks. By comparing the performance scores at 100K and 500K steps of CUDC against the best baseline method, CUDC has achieved on average **18.2% and 12.3% improvement** at respectively 100K and 500K learning steps, across 4 downstream tasks. In particular, our proposed CUDC is **21.1% higher** than the best baseline of APT at 100K steps on Run task while its learning performance is **15.2% higher** than the best baseline of APS at 500K steps.
>
> ### References
> [b] Kumar, Aviral, et al. "Conservative q-learning for offline reinforcement learning." Advances in Neural Information Processing Systems 33 (2020): 1179-1191.
>
> [c] Denis Yarats, David Brandfonbrener, Hao Liu, Michael Laskin, Pieter Abbeel, Alessandro Lazaric, and Lerrel Pinto. Don’t change the algorithm, change the data: Exploratory data for offline reinforcement learning. arXiv preprint arXiv:2201.13425, 2022.

---

> ### Author Response · Authors · 2022-11-18
> **Response to Reviewer bF1X (3/3)**
>
> ### Q4.3 If the main contribution is a reweighting scheme for offline RL, then can that be studied independently of the self-supervised exploration setting?
> **A4.3**: No, the main contribution is an adaptive k-step reachability analysis to automatically determine how many steps into the future that the agent should predict during the data collection process. It avoids fixing a specific step to limit the feature space, and it cannot be concluded as a reweighting scheme for offline RL. In the last paragraph of Section 4, we investigate adapting the k-step in different ways. In particular, we compare two methods of uniform adaptation (Uniform) and gaussian adaptation (Normal).
> 1) Uniform does not rely on self-supervised settings to vary the k-step and it performs much worse than our proposed CUDC. Figure 4 shows that Uniform only achieves **81%** of sample efficiency as CUDC at 100K steps and **85%** performance scores as CUDC at 500K steps, across 4 downstream tasks in Walker. As for the learning stability, Uniform has around **200%** and **300%** of standard deviation as CUDC at respectively 100K and 500K steps.
> 2) As shown in Figure 4, Normal even outperforms our proposed method in the Flip task. However, its overall performance across all 4 downstream tasks is still **4.5% weaker** than ours and its standard deviation is **128% higher** than CUDC, indicating the instability issue caused by this adaptation method.
>
> Based on these results, we strongly believe the proposed self-supervised k-step adaptation is indeed necessary and effective, which is the first-of-its-kind to the best of our knowledge.

---

### Author Response · Authors · 2022-11-19
**Revision Summary**

We would like to thank the reviewers for their valuable comments. We respond to the individual reviews below. Based on reviewer’s suggestions and concerns, we’ve also updated the paper with a number of modifications as follows:

1) We added additional experiments using another offline RL algorithm CQL to further verify the effectiveness of our proposed method in Appendix C.2.
2) We added additional ablation studies by removing each proposed component to demonstrate their individual effect on the stated improvement in Appendix C.4.
3) We elaborated the intuition of constructing the reachability loss in Section 3.2.
4) We added more clarification of k-step exploration in Abstract, Section 1 and Section 2.
5) We added Figure 9 and discussions in Appendix C.1 on how the losses of dynamics model and reachable network change during learning.
6) We added Figure 10 and discussions in Appendix C.1 to visualize the true reward density of the collected dataset on the downstream task.
7) We added a sweep of the upper bound of $k$-step range in Appendix C.5.
8) We added error bars for all bar plots.
9) We updated the term “environment step” by “temporal distance” to avoid confusion.

---

### Comment · Area_Chair_6WzC · 2022-11-27
**Confusion around data efficiency vs. computational efficiency**

Dear authors,

I took a look at the paper myself to contextualize the reviews and your response to the reviews, and I am starting to deliberate with the reviewers. When I was looking at the paper, I noticed a confusing issue in the paper that was not brought up in the reviews, which I would like to raise to you to give you an opportunity to reply.

In particular, the Figure 2 caption describes the figure as showing learning curves of an offline RL agent, and the text also seems to imply that the plot is showing performance over the course of offline RL training. Hence, I would expect the x-axis of the plot to correspond to the number of gradient steps or training iterations. However, the x-axis is labeled as environment steps and the caption claims that the CUDC algorithm improves "sample efficiency." The results in Table 1 also seem to have the same issue -- the numbers match the learning curves, but the caption of Table 1 claims better "sample efficiency."

If the number of steps actually corresponds to gradient steps, there are two issues: (1) the paper has incorrect claims -- the claims should be about computational efficiency not sample efficiency, and (2) computational efficiency is usually much less important than sample efficiency (since, e.g., it is far more expensive to collect data on a real robot than it is to compute a gradient on a computer, and 500k gradient steps is generally quite cheap).

On the other hand, if the number of steps does correspond to environment steps, can you explain how the plot in Figure 2 was sampled so densely? Do you only do one epoch of training over the data? If only one epoch is done, then why not do more, as is typical in offline RL training? And if only one epoch is done, does training proceed step-by-step in the order that the data was collected, or is the data shuffled before training? If it is the latter, then this won't necessarily translate into the actual amount of data that would have to be collected on the system, since the full amount was collected before shuffling and starting training, again invalidating claims about data efficiency. If it is the former, then I would expect better performance with shuffling.

Lastly, I should also note that the paper decision is not necessarily stemming on the fate of this particular issue (e.g. it still could be rejected even if this is resolved) because the reviewer discussion is still on-going. But, again, I wanted to give you an opportunity to reply.

Thank you!

---

> ### Author Response · Authors · 2022-11-29
> **Response to Area Chair 6WzC**
>
> Dear Area Chair 6WzC,
>
> Thank you for your interest and raising your concerns about sample efficiency in our work. We really appreciate it. The number of steps in Figure 2 corresponds to the training steps (gradient steps) of the offline RL agent. We would like to clarify it and address your concerns in the following responses.
>
> ### Q: The paper has incorrect claims -- the claims should be about computational efficiency not sample efficiency. Computational efficiency is usually much less important than sample efficiency
>
> **A**: In online RL, sample efficiency and computational efficiency is clearly distinguishable based on whether new samples are collected through additional interaction with environments. However, sample efficiency in offline RL can be reflected by computational efficiency, since the dataset is previously collected and no interaction with environments is allowed. As summarized in [a] and [b], being sample efficient in offline RL means to distinguish and extract useful samples from the fixed dataset as much as possible such that offline RL agents can learn an optimal / near-optimal policy ideally in a computationally fast way (e.g. with few gradient steps). In Section 1.1 of [b], it shows that sample efficient offline RL is an important and on-going research field even though there is no expensive interaction with the environment involved. The similar definitions can also be found in other offline RL works [c,d,e], where the offline RL methods are claimed to be sample efficient when using fewer data or gradient steps for training while achieving desirable learning performances. Therefore, we followed the same terminology of sample efficiency in our work, which is closely reflected by the computational efficiency. More specifically, training the offline RL agent with 100K gradient steps requires much fewer samples to be drawn from the fixed dataset compared to training with 500K gradient steps, and the performance at 100K gradient steps is used to compare the sample efficiency of the offline RL agent on different collected datasets.
>
>
> In addition, the claim of being sample efficient for our method can be further verified from the following two perspectives.
>
> 1) In our main experiments, we have fixed the collected dataset size to be 1M for different data collection methods. Despite the same amount of data being sampled during each downstream task learning, please kindly note that the offline RL performances are not the same and our proposed method outperforms the baseline methods at both 100K and 500K learning steps for most downstream tasks. Thus, we can conclude that not all collected samples are useful for the downstream tasks and the sample efficiency can also be reflected by the proportion of useful samples in the fixed dataset. From this perspective, our method is indeed sample efficient.
>
> 2) We agree that it may be more expensive to interact with the environment and collect data than to train the offline RL agent with more steps in many scenarios. For this reason, our main motivation is to re-use the collected data across many downstream tasks, which requires the collected data to be task-agnostic as well as useful for offline learning. This problem setting can dramatically improve the sample efficiency, compared to the conventional offline RL setting where offline RL algorithms are only evaluated (trained) on the same task that the dataset was created with. According to our results in Figure 2 and Table 1, our method outperforms the baseline methods in the sense that the offline RL agent can learn multiple downstream tasks well with a fixed task-agnostic dataset.
>
> In our next revision, we will update the x-axis label from “Environment Steps (K)” to “Offline RL Training Steps (K)” for Figure 2 (as well as other offline RL learning curve plots) to be more clear. Many thanks for raising this issue and giving us this opportunity to reply. We would like to expand the discussion and further address any concern you may have.
>
>
>
>
> [a] Shi, Laixi, et al. "Pessimistic q-learning for offline reinforcement learning: Towards optimal sample complexity." arXiv preprint arXiv:2202.13890 (2022).
>
> [b] Xie, Tengyang, et al. "Policy finetuning: Bridging sample-efficient offline and online reinforcement learning." Advances in neural information processing systems 34 (2021): 27395-27407.
>
> [c] Li, Gen, et al. "Settling the sample complexity of model-based offline reinforcement learning." arXiv preprint arXiv:2204.05275 (2022).
>
> [d] Foster, Dylan J., et al. "Offline reinforcement learning: Fundamental barriers for value function approximation." arXiv preprint arXiv:2111.10919 (2021).
>
> [e] Zhang, Longfei, et al. "ORAD: a new framework of offline Reinforcement Learning with Q-value regularization." Evolutionary Intelligence (2022): 1-9.

---

> > ### Comment · Area_Chair_6WzC · 2022-11-29
> > **Reply**
> >
> > Can you clarify if you are arguing about the definition of sample efficiency or about the importance of computational efficiency?
> >
> > The widely used definition of a "sample" is a datapoint, and not a gradient step. Thus, sample efficiency / data efficiency corresponds to the amount of data needed to achieve a certain level of performance. Therefore, if one wanted to make an unambiguous statement or claim about the sample efficiency of an offline RL algorithm, one would need to evaluate performance with different sized datasets.

---

> > > ### Author Response · Authors · 2022-11-30
> > > **Response to Area Chair 6WzC - About Sample Efficiency**
> > >
> > > Dear Area Chair 6WzC,
> > >
> > > In our last reply, we wanted to redefine the sample efficiency in our work from three perspectives: 1) computational efficiency, 2) proportion of useful data in the fixed dataset for offline learning, and 3) reusability of the task-agnostic dataset across many downstream tasks.
> > >
> > > 1) **Computational efficiency**. We agree with your definition of sample efficiency that it should be essentially based on different amounts of data points, but not a gradient step. However, the conventional offline RL settings generally assume the dataset is collected beforehand and fixed throughout offline RL learning. Existing sample efficient offline RL literature aims to learn an optimal / near-optimal policy by extracting useful samples from the fixed dataset or in a computationally fast way (e.g. with few gradient steps). Thus, we were using computational efficiency at 100K offline RL training steps to reflect the sample efficiency in offline RL to some extent, as the amount of 100K batches of sampled data from the fixed dataset is smaller than that of 500K batches of sampled data. In addition, both Figure 2 and Table 1 shows that the performance score in the Walker-Stand task at 100K gradient steps is higher than that of 500K gradient steps, and the performance scores in the Walker-Run and Quadruped-Stand tasks at 100K gradient steps are close to those at 500K gradient steps, indicating desirable computational efficiency.
> > >
> > > However, we realize that computational efficiency alone is not sufficient to claim the sample efficiency of our method. Thus, we would like to future clarify it with the additional two perspectives.
> > >
> > > 2) **Proportion of useful data in the fixed dataset for offline learning**. As shown in Figure 2, despite the same amount of data being sampled at each gradient step, our proposed method outperforms the baseline methods at both 100K and 500K learning steps in most downstream tasks. Thus, we observe that not all collected samples are useful for the offline RL agent and our method is sample efficient in the sense that it collects higher-quality dataset with a larger proportion of useful data points in the fixed dataset for the downstream tasks. We have visualized this in Figure 10 by plotting the true reward distribution of the collected data points on a downstream task.
> > >
> > > 3) **Reusability of the task-agnostic dataset**. Our main motivation is to re-use the collected dataset across many downstream tasks, which avoids the expensive data collection for each individual downstream task and thereby improves the sample efficiency. This problem setting is emerging and of great significance for the offline RL research community. According to our results in Figure 2 and Table 1, our method outperforms the baseline methods in the sense that the offline RL agent can perform multiple downstream tasks well with a fixed task-agnostic dataset.
> > >
> > > We will update the claim of sample efficiency carefully in the next revision, not only just based on the computational efficiency at 100K gradient steps, but also by including the justification of Point 2 and 3 above. Many thanks for highlighting this issue! We look forward to addressing any further concern that you or the reviewers may have.

---

### Decision · Program_Chairs · 2023-01-20

**Decision:**

Reject

**Justification For Why Not Higher Score:**

After the author response, he main reviewer criticism stands: the method is a somewhat arbitrary combination of many established exploration objectives, with limited performance improvement for the significant added complexity. That is, the improvements over prior methods are somewhat small with many of the error bars overlapping, and the proposed method is significantly more complex than these prior methods, so the increase in performance does not seem to warrant the added complexity. There are also issues with the presentation of the evaluation that need to be fixed, i.e. to account for the important distinction between sample efficiency and computational efficiency, which are conflated in the current paper.

**Justification For Why Not Lower Score:**

n/a

**Metareview: Summary, Strengths And Weaknesses:**

**Strengths**: The reviewers agree that this paper studies an interesting and important problem. The results are also promising, and some reviewers found the clarity to be good. The author response addressed some of the concerns.

**Weaknesses**: After the author response, he main reviewer criticism stands: the method is a somewhat arbitrary combination of many established exploration objectives, with limited performance improvement for the significant added complexity. That is, the improvements over prior methods are somewhat small with many of the error bars overlapping, and the proposed method is significantly more complex than these prior methods, so the increase in performance does not seem to warrant the added complexity. There are also issues with the presentation of the evaluation that need to be fixed, i.e. to account for the important distinction between sample efficiency and computational efficiency, which are conflated in the current paper.

Overall, the weaknesses outweigh the strengths.